# Inhibitory peptidergic modulation of *C. elegans* serotonin neurons is gated by T-type calcium channels

Kara E Zang, Elver Ho, Niels Ringstad*

Skirball Institute for Biomolecular Medicine, The Helen L. and Martin S. Kimmel Center for Biology and Medicine, Department of Cell Biology, NYU Langone School of Medicine, New York, United States

**Abstract** Serotonin is an evolutionarily ancient molecule that functions in generating and modulating many behavioral states. Although much is known about how serotonin acts on its cellular targets, how serotonin release is regulated *in vivo* remains poorly understood. In the nematode *C. elegans*, serotonin neurons that drive female reproductive behavior are directly modulated by inhibitory neuropeptides. Here, we report the isolation of mutants in which inhibitory neuropeptides fail to properly modulate serotonin neurons and the behavior they mediate. The corresponding mutations affect the T-type calcium channel CCA-1 and symmetrically re-tune its voltage-dependencies of activation and inactivation towards more hyperpolarized potentials. This shift in voltage dependency strongly and specifically bypasses the behavioral and cell physiological effects of peptidergic inhibition on serotonin neurons. Our results indicate that T-type calcium channels are critical regulators of a *C. elegans* serotonergic circuit and demonstrate a mechanism in which T-type channels functionally gate inhibitory modulation *in vivo*.

**\*For correspondence:** Niels. Ringstad@med.nyu.edu

**Competing interests:** The authors declare that no competing interests exist.

## Introduction

Neuromodulators constitute a class of diverse chemical signals that can cause widespread and long-lasting effects on neural circuits. Many studies of vertebrate and invertebrate nervous systems illustrate how the combined impact of multiple neuromodulators on brain circuits can generate a recognizable behavioral state, suggesting that the regulation of neuromodulator release is critical for the generation and control of behavior *in vivo*. In the vertebrate central nervous system, the neuromodulator serotonin is released by a small population of neurons that project widely and exert influence on neural circuits throughout the brain. Serotonin is required for central control of basic physiological functions such as thermoregulation and respiratory control (*Myers, 1981*; *Morin et al., 1990*; *Frazer and Hensler, 1999*). Serotonin also functions in the control of affective states and behavioral drives. For example, serotonergic systems control aggression (*Kravitz, 1988*), appetite (*Lent, 1985*; *Hatcher et al., 2008*; *Oh et al., 2015*), and sexual behavior (*Uphouse, 2014*). Dysregulation of endogenous serotonin systems is thought to underlie psychiatric illness, notably affective disorders such as major depression and generalized anxiety (*Holmes et al., 2003*; *Neumeister et al., 2004*). Because serotonin plays critical roles in many different neural circuits that influence affect, cognition, and behavior, components of serotonin signaling pathways are important therapeutic targets for the treatment of neuropsychiatric disorders (*Celada et al., 2013a*, *2013b*; *Angoa-Pérez and Kuhn, 2015*). Current targets include serotonin receptors, serotonin transporters that mediate vesicular packaging or reuptake after release, and catabolic enzymes that degrade serotonin (*Celada et al., 2013a*). Mechanisms that control serotonin release *in vivo* could be similarly important therapeutic targets. However, little is known about these mechanisms.

A simple behavior of the nematode worm *C. elegans* offers a powerful model for the study of molecular mechanisms that regulate serotonin circuits *in vivo*. Female reproductive behavior - egg laying - requires serotonin, which is released from a pair of Hermaphrodite-Specific Neurons (HSNs) (*Trent et al., 1983*; *Desai and Horvitz, 1989*). Egg-laying behavior is highly stereotyped, and when it is disrupted animals visibly bloat with unlaid eggs. This easily identified behavioral change makes the neurochemical systems that generate and modulate egg-laying behavior amenable to genetic analysis. Genes that function in serotonin synthesis, release, and receptor signaling mechanisms mutate to cause egg-laying defects, and unbiased genetic screens for behavioral mutants defective in egg-laying behavior have identified genes and molecules that regulate the function of serotonin neurons (*Trent et al., 1983*; *Desai and Horvitz, 1989*; *Ségalat et al., 1995*; *Mendel et al., 1995*; *Brundage et al., 1996*; *Koelle and Horvitz, 1996*).

The egg-laying circuit comprises three cell-types: the serotonergic HSNs, the cholinergic ventral cord neurons VC4 and VC5, and electrically coupled vulval and uterine muscles, of which the vulval muscles (VMs) receive direct synaptic input from motor neurons (*Figure 1—figure supplement 1*). HSNs synapse both onto VM muscle targets and onto VC4 and VC5 motor neurons. VC4 and VC5 motor neurons synapse onto VMs (*White et al., 1986*) and also provide extrasynaptic inhibitory modulation to the HSNs (*Bany et al., 2003*; *Shyn et al., 2003*). In addition to modulation from local neurons, HSNs also receive inhibitory peptidergic inputs from multiple sources. The G protein-coupled neuropeptide receptor EGL-6 is expressed by HSNs and mediates their inhibition in response to FLP-10 and FLP-17 neuropeptides. FLP-10 inhibitory neuropeptides are secreted from several neuron types, as well as from epithelial cells of the reproductive system. FLP-17 peptides are secreted by a small number of cells and are principally released by a pair of chemosensory neurons, the BAGs (*Ringstad and Horvitz, 2008*; *Kim and Li, 2004*). Both inhibitory cholinergic and peptidergic modulation of the HSNs are mediated by the $G_o$ alpha ortholog GOA-1, which acts through two effectors to mediate neuronal inhibition: the diacylglycerol kinase DGK-1 and inward rectifying potassium channels comprising IRK-1 subunits (*Nurrish et al., 1999*; *Emtage et al., 2012*).

A point mutation in the neuropeptide receptor EGL-6 causes it to transduce excess inhibitory signaling and thereby inactivates the HSNs. Inhibition of HSNs by this mutant receptor requires its endogenous ligands (*Ringstad and Horvitz, 2008*), indicating that EGL-6(gf) receptors are not constitutively active but rather amplify peptidergic inhibition received by HSNs. To identify factors required for inhibitory peptidergic modulation of the HSNs, we sought mutants that were suppressed for the egg-laying defects caused by *egl-6(gf)*. Here, we report that *cca-1*, which encodes a voltage-gated calcium channel homologous to vertebrate T-type calcium channels, mutates to strongly and specifically suppress the effects of excess peptidergic inhibition onto serotonin neurons. The *cca-1* suppressor mutation symmetrically shifts the voltage dependencies of channel activation and inactivation, resulting in a T-type channel that operates over more hyperpolarized membrane potentials compared to wild-type channels. Re-tuning the voltage dependency of T-type channels restores activity to inhibited HSNs at the cellular level and, at the whole animal level, restores wild-type reproductive behavior to mutants in which HSNs receive excess peptidergic inhibition. We propose that these data reveal a general mechanism by which T-type calcium channels act as functional gates to enable or disable inhibitory neuromodulatory input to cellular targets and circuits.

## Results

### *n5209* is a potent suppressor of behavioral effects caused by increased peptidergic inhibition

A mutation in the neuropeptide receptor EGL-6 increases inhibitory signaling to the HSNs in a manner that requires endogenous neuropeptide ligands and $G_o$ signaling (*Ringstad and Horvitz, 2008*). As a consequence of increased inhibition onto the HSNs, *egl-6(gf)* mutants bloat with unlaid eggs. To identify factors required for modulation of the HSNs by inhibitory neuropeptides, we performed a chemical mutagenesis screen for suppressors of the *egl-6(gf)* behavioral phenotype. Our screen yielded six potent suppressors, including the suppressor mutation *n5209*. *egl-6(gf)* mutants that also carry the *n5209* suppressor mutation are not bloated with unlaid eggs (*Figure 1A*). To exclude the possibility that *n5209* mutants retain fewer eggs solely because they produce fewer eggs, we assayed their egg-laying behavior by measuring the developmental stage of newly released

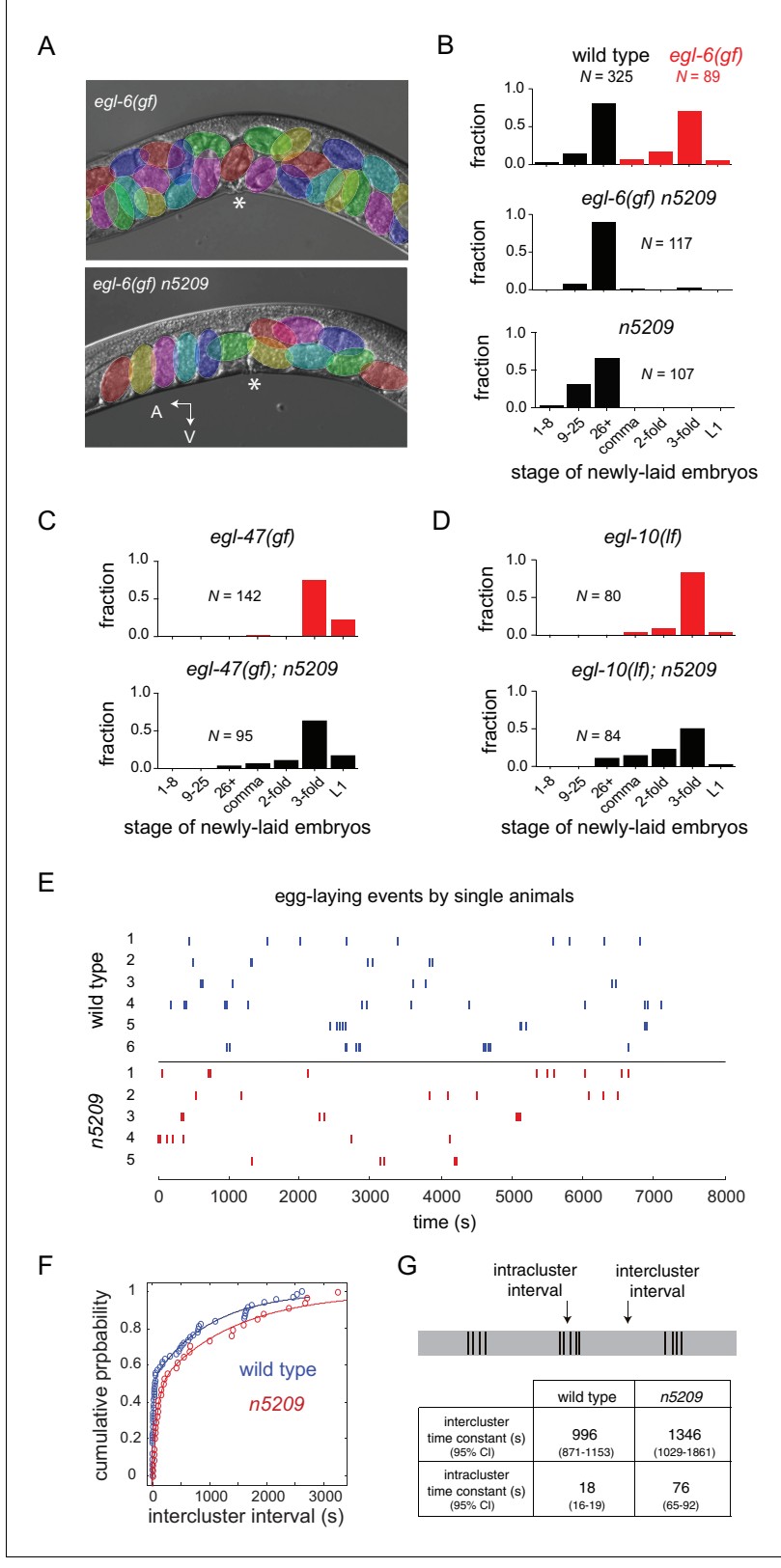

**Figure 1.** The *n5209* mutation is a potent and specific suppressor of inhibitory neuropeptide signaling. (**A**) Micrograph of *egl-6(gf)* and *egl-6(gf) n5209* animals. Eggs are colored for emphasis. Asterisk indicates vulva. (**B**) Distribution of the developmental stages of eggs laid by wild type, *egl-6(gf)*, *egl-6(gf) n5209*, and *n5209* animals. **C** and **D**) *n5209* has little effect on *egl-47(gf)* or *egl-10(lf)*. Shown are distributions of the developmental stages of

*Figure 1 continued*

eggs laid by *egl-47(gf)* and *egl-10(lf)* mutants with and without the *n5209* allele. (E) Raster plots of egg-laying events for wild type animals (blue) and *n5209* animals (red). Single animals were tracked for two hours. Each egg-laying event is represented by a hash mark. (F) Cumulative probability plots of wild type (blue) and *n5209* (red) intercluster intervals fit to the model described by *Waggoner et al. (1998)*. (G) Time constants computed from the same model. 95% confidence interval is in parentheses. Connectivity of the egg-laying neuromusculature is depicted in *Figure 1—figure supplement 1*, and the data from 1F are replotted as a histogram of the natural log of interval times in *Figure 1—figure supplement 2*.

The following source data and figure supplements are available for figure 1:

**Source data 1.** Raw data of developmental stage of newly laid embryos from *Figure 1*.

**Source data 2.** Time of egg-laying event during tracking.

**Figure supplement 1.** Anatomy and connectivity of the *C. elegans* egg-laying circuit.

**Figure supplement 2.** Histogram of interval times on a natural log scale.

embryos. This assay uses embryonic development as a clock to measure how long embryos have been held *in utero* and is independent of the rate of egg production. Consistent with their having reduced rates of egg laying, *egl-6(gf)* mutants released late-stage embryos. By contrast, *egl-6(gf) n5209* mutants released earlier stage embryos that had not yet undergone extensive morphogenesis, similar to wild type animals (*Figure 1B*; *P*-values for all noted comparisons are in *Supplementary file 1*).

Inhibition of HSNs by *egl-6(gf)* is mediated by increased G$_o$ signaling downstream of the activated neuropeptide receptor (*Ringstad and Horvitz, 2008*). Other mutations also activate G$_o$ signaling in HSNs to cause egg-laying defects. Specifically, loss-of-function mutations affecting the GTPase-activating protein EGL-10, which is a negative regulator of G$_o$ in HSNs, and an activating mutation in the orphan G protein-coupled receptor EGL-47 cause profound defects in HSN function that strongly depend on G$_o$ signaling (*Koelle and Horvitz, 1996*; *Moresco and Koelle, 2004*). Previously, we found that the inward rectifier potassium channel gene *irk-1* was required for inhibition of HSNs by EGL-6 but was not generally required for inhibitory G$_o$ signaling in HSNs (*Emtage et al., 2012*). We tested whether the *n5209* suppressor mutation displays a similar specificity for signaling pathways downstream of the neuropeptide receptor EGL-6. We found that *n5209* strongly suppressed the *egl-6(gf)* phenotype but had little effect on the egg-laying defects of *egl-47(gf)* and *egl-10(lf)* mutants (*Figure 1C and D*). These data suggest that the gene affected by *n5209*, like *irk-1*, is not generally required for inhibitory G$_o$ signaling in HSNs but has a privileged function with respect to signaling downstream of receptors for inhibitory neuropeptides.

## *n5209* does not change the temporal structure of female reproductive behavior

We considered the possibility that the *n5209* mutation might constitutively activate the reproductive neuromusculature and bypass its normal regulation. To test this hypothesis, we separated the *n5209* suppressor mutation from the *egl-6(gf)* mutation and determined whether *n5209* mutants had abnormal reproductive behavior. We observed that *n5209* alone had no measurable effect on the developmental stage of newly laid eggs (*Figure 1B*, bottom). These data suggest that *n5209* does not grossly affect the timing of egg-laying behavior. We next used high-resolution behavior analysis to quantify parameters of egg-laying behavior of *n5209* mutants. *C. elegans* egg-laying behavior is a stochastic process that is well described by a model in which the neuromusculature switches between active and inactive states with dynamics dictated by measurable rate constants (*Waggoner et al., 1998*). We tracked animals carrying the *n5209* mutation to determine whether *n5209* affected state transitions in the egg-laying system and, if so, whether the rate constants determining these transitions were significantly altered by the suppressor mutation. We observed that the behavior of *n5209* mutants, like that of the wild type, switched between active and inactive states

and displayed a very similar temporal structure (*Figure 1E*). We noted that one parameter of the model used to describe the behavior is altered in *n5209* mutants: the time constant that defines the rate with which the egg-laying system 'resets' after a behavioral event (*Figure 1F and G* and *Figure 1—figure supplement 2*). As a result, *n5209* mutants display slightly longer intervals between egg-laying events when the system is in an active state. We noted that *n5209* animals carry slightly fewer embryos *in utero* than do wild-type animals (average of 7.3 ± 2.3 embryos vs 11.25 ± 3.4 embryos, respectively). This might explain the slight increase in the intracluster time constant. However, the overall structure of the reproductive behavioral program is intact in *n5209* mutants. These data indicate that the *n5209* mutation restores behavior to *egl-6(gf)* mutants and does not simply cause unregulated activation of the reproductive neuromusculature.

## *n5209* affects the T-type calcium channel CCA-1

Using three-factor crosses, high-resolution SNP mapping, and whole-genome sequencing, we initially mapped the *n5209* suppressor to the X chromosome within an interval that contained four mutations predicted to affect protein-coding sequences (*Figure 2A*). We separated these mutations by recombination and found that recombinants with a mutation in *cca-1* were always suppressed for egg-laying defects (4/4 recombinants) while recombinants that lost this mutation were never suppressed (5/5 recombinants). The *cca-1* mutation is a C->T base change in the seventh exon (*Figure 2A*). *cca-1* encodes the pore-forming subunit of a low-voltage activated (LVA) T-type calcium channel and is the only T-type calcium channel gene encoded by the *C. elegans* genome (*Shtonda and Avery, 2005*; *Steger et al., 2005*). We mapped a second independently derived suppressor mutation, *n6122*, to the same linkage group and found that *n6122* mutants carry the identical sequence change in *cca-1* (*Figure 2A* and *Figure 2—figure supplement 1*). That the *n5209* and *n6122* mutations were independently derived is indicated by the large number of unique sequence variants near the *cca-1* locus in each of the mutants (*Figure 2—figure supplement 1*). Because we mapped the *n5209* mutation to a small interval that contains only one mutation that affects protein-coding sequence and because the identical mutation was recovered independently by our screen, we concluded that the *n5209* and *n6122* suppressor mutations affect the T-type calcium channel gene *cca-1*.

*n5209* and *n6122* are predicted to affect a proline residue in a short cytoplasmic loop that connects the voltage-sensing domain (transmembrane domain 4) to transmembrane domain 5 (*Figure 2B*). This linker is highly conserved among T-type calcium channels from diverse animal phyla (*Senatore et al., 2014*) and the proline affected by the *n5209* and *n6122* suppressor mutation is invariant between vertebrates and nematodes, suggesting its functional importance (*Figure 2C*).

We noted that *cca-1(n5209)* is a semi-dominant suppressor of the *egl-6(gf)* phenotype (*Figure 3A*). Also, a deletion allele of *cca-1 - ad1650 -* did not suppress *egl-6(gf)* as well as did *cca-1(n5209)* (*Figure 3B*). Furthermore, *cca-1(ad1650)* caused an egg-laying defect (*Figure 3C*), whereas *cca-1(n5209)* mutants exhibited wild-type egg-laying behavior (*Figure 1B*). These data indicate that *n5209* does not simply cause loss of *cca-1* function but rather alters the function of the CCA-1 T-type calcium channel to restore behavior to *egl-6(gf)* mutants.

## *n5209* alters the voltage dependency of the voltage-gated T-type calcium channel

To determine the effect of the suppressor mutation on CCA-1 function, we sought to record currents passed by wild-type and mutant channels in a heterologous system. We were unable to express the *C. elegans* CCA-1 channels in a heterologous system, so we introduced the corresponding mutation into a very highly conserved mammalian homolog of CCA-1, Ca$_v$3.1(*Figure 3C*). Ca$_v$3.1 is readily expressed in heterologous expression systems such as *X. laevis* oocytes (*Cribbs et al., 1998*; *Perez-Reyes et al., 1998*). We recorded whole-cell currents from transgenic oocytes expressing either wild-type channels or channels carrying the suppressor mutation. Voltage steps used to measure channel activation and representative traces for wild-type and mutant channels are shown in *Figure 4A and B*, respectively. We observed that the P262S suppressor mutation shifted the current-voltage relationship of T-type channels towards more hyperpolarized potentials (*Figure 4—figure supplement 1*). The suppressor mutation shifted both the steady-state activation and inactivation curves. The peak conductance was shifted by −10 mV, with wild-type channels exhibiting

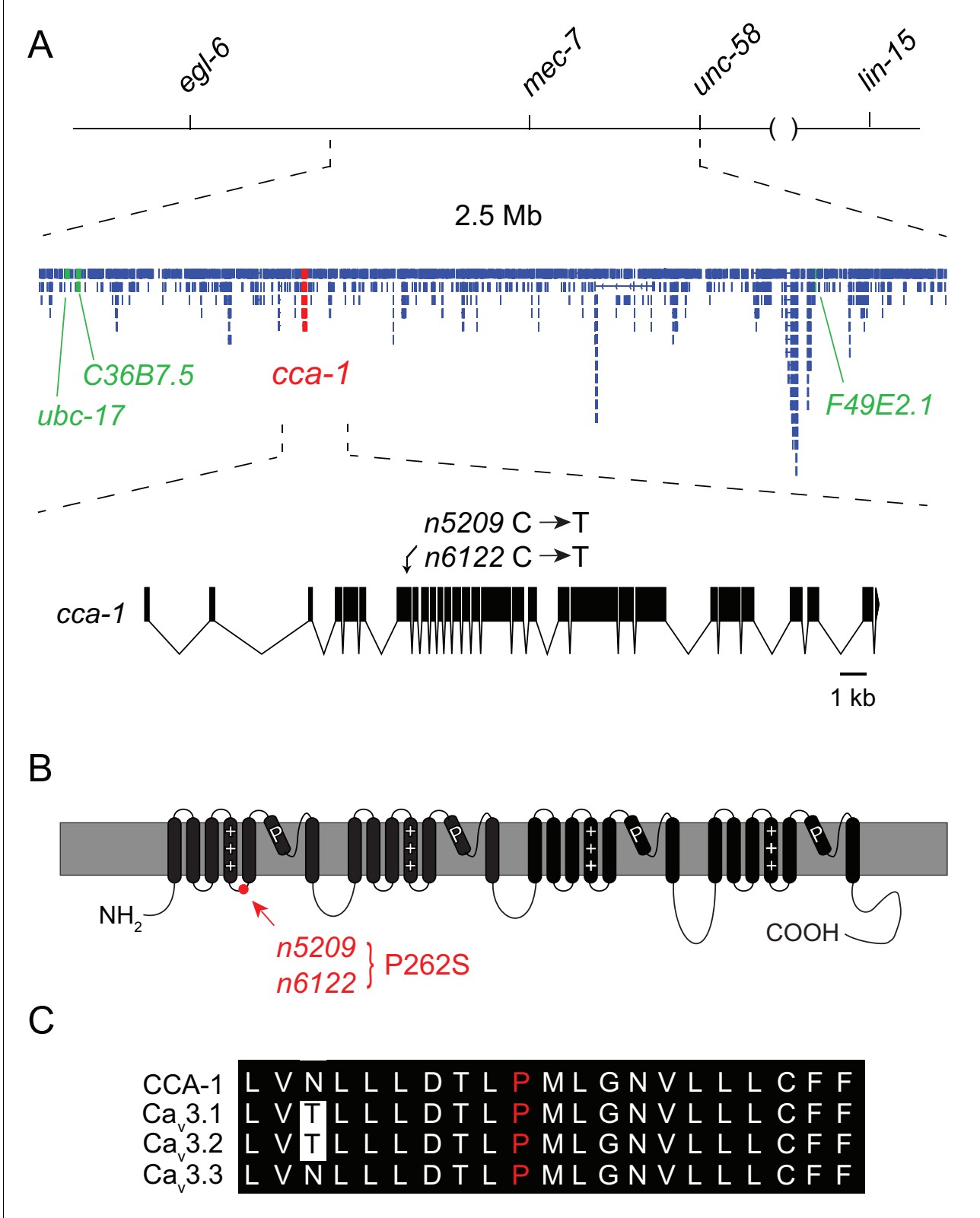

**Figure 2.** The *n5209* suppressor of peptidergic inhibition affects the T-type calcium channel gene *cca-1*. (A) The suppressor mutation, *n5209*, was mapped to a 2.5 Mb interval on Linkage Group X using a marked *egl-6(gf)* strain (top). Inset: Gene models within the 2.5 Mb interval. Sequence variants in this interval predicted to affect protein-coding sequences and that were used as markers for genetic mapping are in green. The causative suppressor mutation in *cca-1* is in red. The *cca-1* gene model is enlarged to show detail (bottom). (B) Schematic of CCA-1 protein. *n5209* and *n6122* are predicted
*Figure 2 continued on next page*

*Figure 2 continued*

to cause a proline to mutate to a serine in the intracellular domain next to the voltage sensor. (**C**) Amino acid sequence alignment shows that the affected proline residue is present in all three mammalian T-type channels. All sequence variants in this interval in *n5209* and *n6122* mutants are shown in *Figure 2—figure supplement 1*.

The following figure supplement is available for figure 2:

**Figure supplement 1.** X-linked sequence variants in *n5209* and *n6122* mutants.

peak conductance at −20 mV and mutant channels exhibiting peak conductance at −30 mV (*Figure 4C*). The half-activation potential was shifted by −7 mV, from −37.14 ± 2.4 mV (wild-type) to −44.69 ± 1.4 mV (mutant). We observed that the suppressor mutation caused a similar shift in the steady-state inactivation curve of T-type channels. Voltage steps used to measure channel inactivation and representative traces for wild-type and mutant channels are shown in *Figure 4D and E*, respectively. The half-inactivation potential shifted from −58.18 ± 0.7 mV (wild-type) to −63.11 ± 0.3 mV (mutant) (*Figure 4F*). The slope factor ($k$) was unchanged for both steady-state activation and inactivation.

T-type channels exhibit a characteristic overlap of the steady-state activation and inactivation curves (*Coulter et al., 1989*; *Crunelli et al., 1989*; *Hernández-Cruz and Pape, 1989*; *Huguenard, 1996*). This overlap gives rise to a small, steady-state 'window current'. These data indicate that mutant channels pass a window current over a range of more hyperpolarized potentials than do wild-type channels, but the width of that range is not significantly affected (*Figure 4G and H*).

## The *cca-1* suppressor mutation functions in parallel to an intracellular signaling pathway that mediates peptidergic inhibition

In some contexts, G protein signaling directly regulates T-type calcium channels (*Sun and Dale, 1997*; *Wolfe et al., 2003*; *DePuy et al., 2006*; *Hu et al., 2009*; *Perez-Reyes, 2010*). To determine whether this is the case in the serotonergic HSNs, we tested whether *cca-1(n5209)* had any effect in the absence of the $G_{i/o}$ ortholog GOA-1 or in the absence of neuropeptide ligands for the GPCR EGL-6. Loss of the sole $G_{i/o}$ ortholog, GOA-1, potently suppresses *egl-6(gf)* (*Ringstad and Horvitz, 2008*). We found that the *cca-1* suppressor mutation enhanced the suppression by loss of *goa-1* function (*Figure 5A*), suggesting that the *cca-1* suppressor mutation has an effect in parallel to G proteins in the control of cellular inhibition. We also found that *cca-1(n5209)* enhanced the suppression of the *egl-6(gf)* phenotype by mutation of the ligand-encoding genes *flp-10* and *flp-17* (*Figure 5B*). However, *cca-1(n5209)* had no measurable effect on animals that lack HSNs (*Figure 5C*). These data indicate that although CCA-1 acts in parallel to EGL-6, it nevertheless depends on the HSNs to modify reproductive behavior (*Figure 5D*).

## T-type calcium channels regulate function of the serotonergic neurons that drive female reproductive behavior

To test the hypothesis that the site of T-type channel function in the egg-laying system is the HSNs, we first sought to determine the cellular expression pattern of CCA-1. We generated a fosmid-based *cca-1::GFP* reporter transgene that contains 22 kb of regulatory sequence 5' to *cca-1* coding sequences. We observed *cca-1::GFP* expression in neurons of the egg-laying system. Both the cholinergic VC4 and VC5 neurons and the serotonergic HSNs strongly expressed *cca-1::GFP* (*Figure 6B*). We noted that a shorter 3 kb fragment of upstream regulatory sequences only drove reporter expression in VC4 and VC5 and not the in the HSNs (*Figure 6—figure supplement 1*). We also observed *cca-1::GFP* expression in many neurons in the head (*Figure 6A*), tail (data not shown), and in the ventral cord. We also observed *cca-1::GFP* expression in the non-neuronal distal tip cells of the somatic gonad (*Figure 6B*).

We next took advantage of an existing mutant that disrupts gene expression in neurons of the egg-laying system to determine where CCA-1(n5209) mutant channels function to suppress the behavior defects of *egl-6(gf)* mutants. LIN-39 is a homeodomain-containing transcription factor that specifies many cell fates in the midbody. LIN-39 is required for the development of VC neurons but

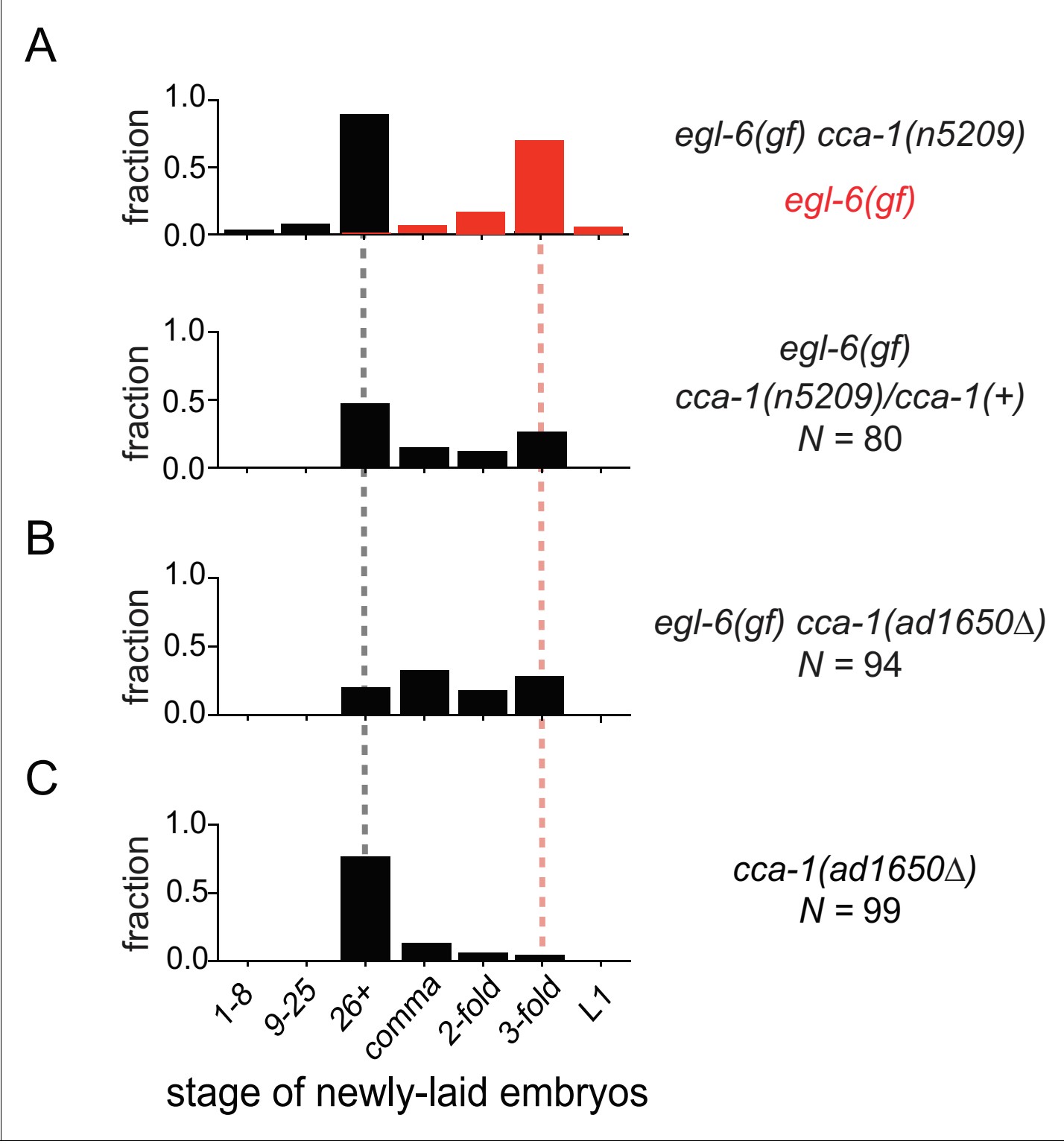

**Figure 3.** The *n5209* suppressor mutation alters *cca-1* function. (**A**) *cca-1(n5209)* is a potent suppressor of *egl-6(gf)* (top; data replotted from *Figure 1*). One copy of the *n5209* allele was sufficient to partially suppress the *egl-6(gf)* phenotype (bottom). (**B**) A deletion allele of *cca-1* partially suppressed *egl-6(gf)*, but not as potently as *n5209* did. (**C**) *cca-1(Δ)* significantly altered the stage of newly laid embryos when compared to wild type.

The following source data is available for figure 3:

**Source data 1.** Raw data of developmental stage of newly laid embryos from *Figure 3*.

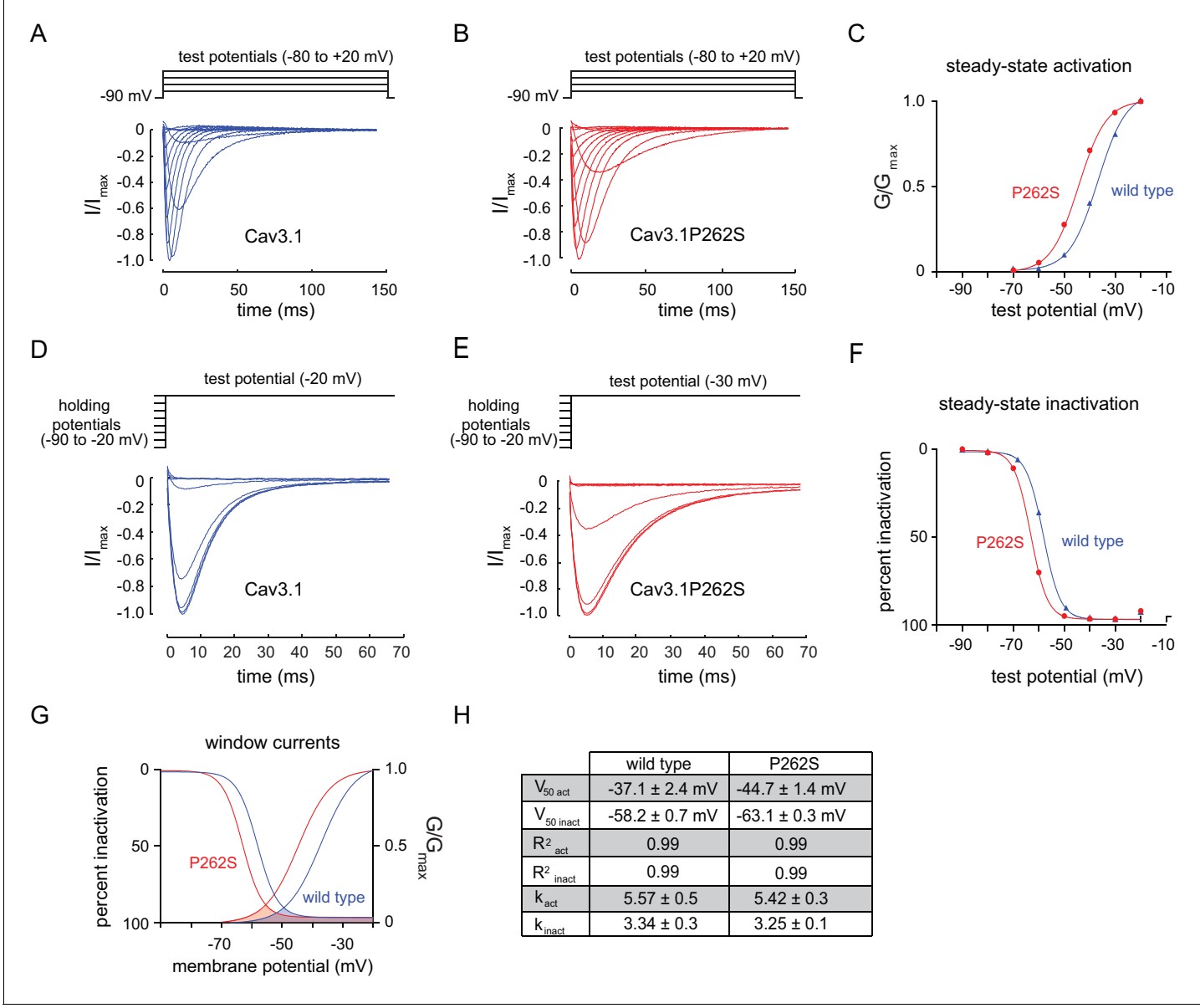

**Figure 4.** The *n5209* mutation changes the voltage-dependence of steady-state activation and inactivation of T-type channels. (**A–C**) Steady-state activation of wild-type and mutant T-type channels expressed in *X. laevis* oocytes. (**A**) Representative trace of wild type $Ca_v3.1$, the mammalian homolog of CCA-1. (**B**) Representative trace of mutant $Ca_v3.1$ ($Ca_v3.1P262S$). (**C**) Steady-state activation curve. Chord conductances were obtained using the Goldman-Hodgkin-Katz equation. $Ca_v3.1P262S$ was activated at significantly more hyperpolarized potentials than wild type. (**D–F**) Steady state inactivation measured in *X. laevis* oocytes. (**D**) Representative trace of wild type $Ca_v3.1$. (**E**) Representative trace of $Ca_v3.1P262S$. (**F**) Steady-state inactivation curve. (**G**) $Ca_v3.1P262S$ had a window current that is significantly shifted toward hyperpolarized potentials. (**H**) Table of values for wild-type and mutant $Ca_v3.1$ channels. *N* = 10 for activation of wild-type $Ca_v3.1$ channels. *N* = 5 for activation of mutant $Ca_v3.1P262S$ channels. *N* = 7 for inactivation of wild-type channels. *N* = 6 for inactivation of mutant channels. For the mutant and wild-type IV curves, see ***Figure 4—figure supplement 1***.

The following figure supplement is available for figure 4:

**Figure supplement 1.** *n5209* mutation shifts the current-voltage relationship of Cav3.1 to more hyperpolarized potentials.

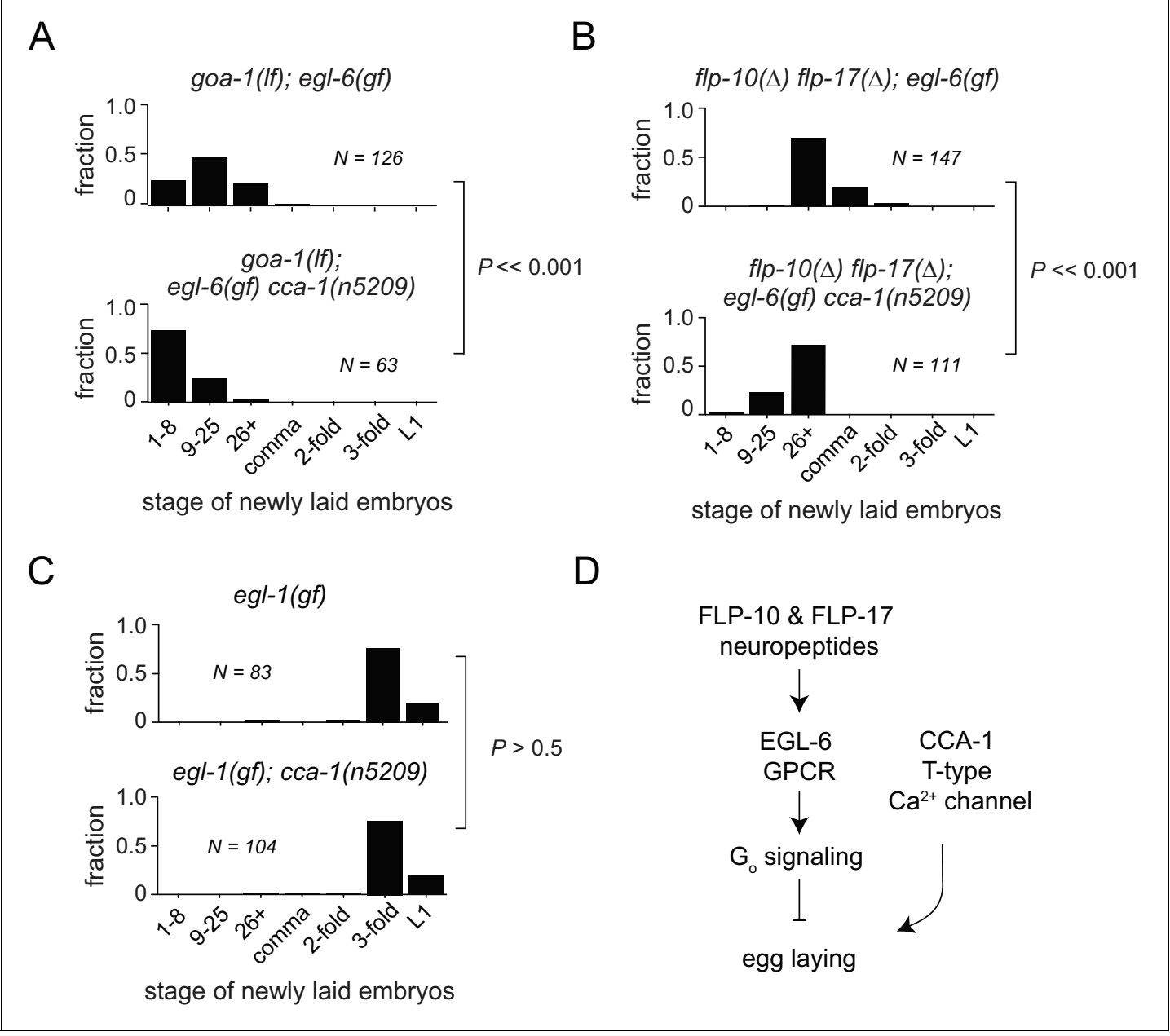

**Figure 5.** T-type Ca$^{2+}$ channels act in parallel to the GPCR signaling pathway that mediates peptidergic inhibition. (**A**) *cca-1(n5209)* enhanced suppression of the *egl-6(gf)* phenotype caused by loss of its downstream effector, GOA-1. (**B**) *cca-1(n5209)* enhanced the suppression of *egl-6(gf)* caused by the loss of its neuropeptide ligands, FLP-10 and FLP-17. (**C**) *cca-1(n5209)* had no effect on animals that do not have HSNs. (**D**) Our data support a model in which CCA-1(n5209) acts in parallel to EGL-6 signaling.

The following source data is available for figure 5:

**Source data 1.** Raw data of developmental stage of newly laid embryos from *Figure 5*.

is not required for the development of HSNs, which are born in the posterior part of the embryo and migrate to their final position in the midbody (*Clark et al., 1993*; *Desai et al., 1988*). We found that LIN-39 was required for *cca-1* expression in VC4 and VC5 neurons (*Figure 6C and D*, and *Figure 6—figure supplement 1*). By contrast, *cca-1* expression in HSNs was not affected by *lin-39* mutation (*Figure 6C*). We next tested whether *lin-39* was required for the ability of *cca-1(n5209)* to

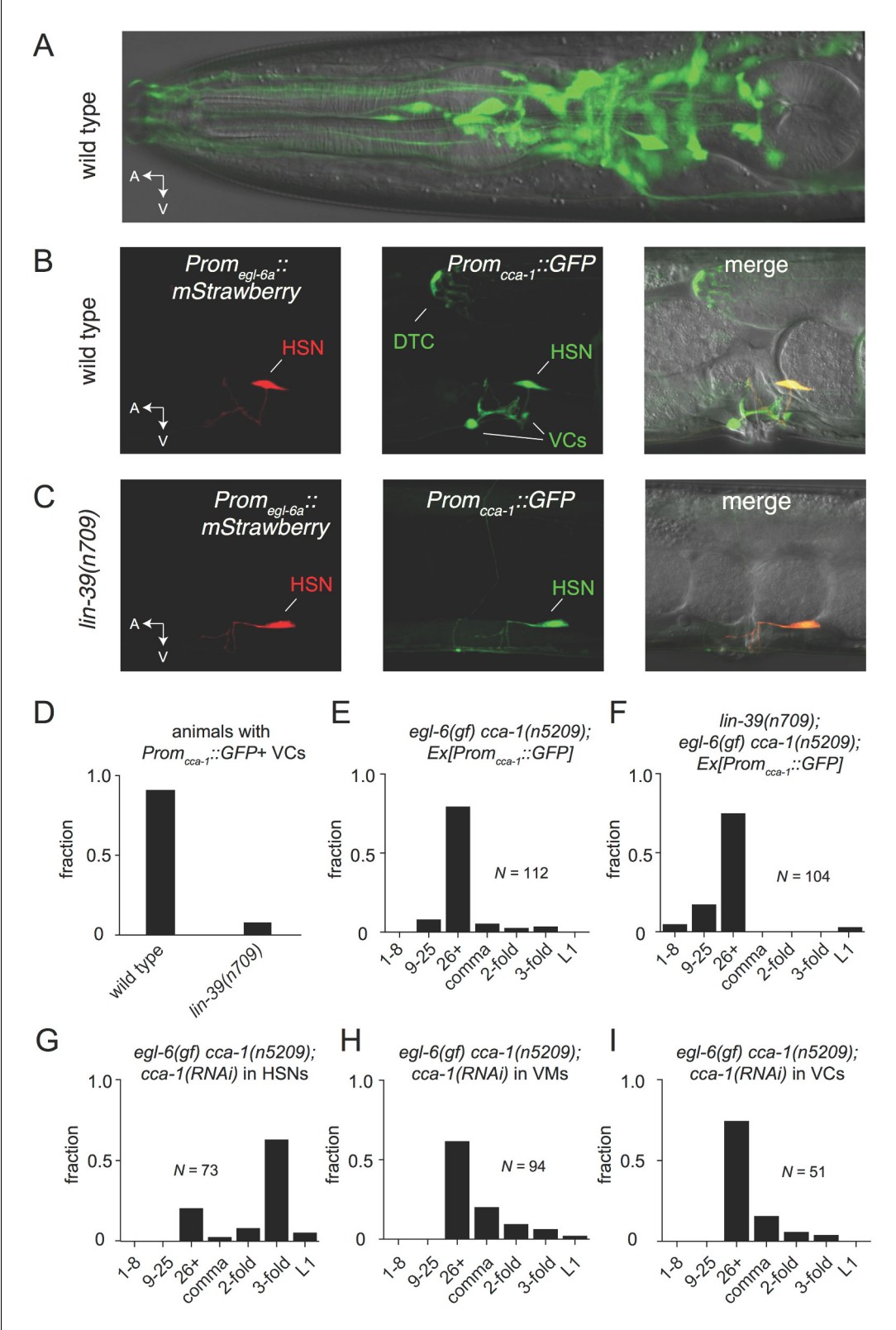

**Figure 6.** The T-type Ca$^{2+}$ channel CCA-1 acts in the HSNs to suppress excess neuropeptide inhibition. (**A**) A *cca-1* fosmid reporter (green) is expressed in many head neurons. (**B**) A *cca-1* fosmid reporter (green) is expressed in the egg-laying system and is co-expressed in the HSNs with an *egl-6* reporter (red). (**C**) Expression of a *cca-1* fosmid reporter in VCs but not HSNs requires the transcription factor LIN-39. (**D**) Fraction of animals expressing *Prom$_{cca-1}$::GFP* in VC4 and VC5. (**E**) Distribution of developmental stages of embyros laid by *egl-6(gf) cca-1(n5209)* animals carrying *Prom$_{cca-}$*

*Figure 6 continued on next page*

*Figure 6 continued*

₁::GFP. (F) Distribution of developmental stages of embyros laid by *lin-39(n709); egl-6(gf) cca-1(n5209)* animals carrying *Prom_cca-1_::GFP*. (G–I) Tissue-specific knockdown of CCA-1 in the egg-laying neuromusculature. Knockdown of *cca-1* in HSNs of *egl-6(gf) cca-1(n5209)* mutants restored the *egl-6(gf)* Egl phenotype. All three tissue-specific constructs utilized the same seed sequence to target *cca-1* (listed in **Supplementary file 5**). Each transgene was tested in multiple independently derived lines. Shown are representative lines. Expression of the short *cca-1* promoter is shown in **Figure 6—figure supplement 1**. HSN-specific knockdown of CCA-1 in a wild-type background is shown in **Figure 6—figure supplement 2**.
The following source data and figure supplements are available for figure 6:

**Source data 1.** Number of GFP+ VCs with the *cca-1* fosmid.
**Source data 2.** Raw data of developmental stage of newly laid embryos from **Figure 6**.
**Figure supplement 1.** CCA-1 expression in the VCs is dependent on the homeodomain-containing transcription factor *lin-39*.
**Figure supplement 1—source data 1.** Number of GFP+ VCs with the short *cca-1* promoter.
**Figure supplement 2.** Knockdown of CCA-1 in wild-type HSNs weakly affects egg-laying.

suppress *egl-6(gf)*. *lin-39* mutation did not strongly modify the suppression of *egl-6(gf)* by *cca-1 (n5209)*, strongly suggesting that *cca-1* expression in VC4 and VC5 is dispensable for its function as a suppressor of peptidergic inhibition (**Figure 6E and F**).

We next tested whether CCA-1(n5209) mutant channels are required in HSNs to suppress the inhibition mediated by EGL-6(gf) receptors. We used the *egl-6a* promoter to drive *cca-1(RNAi)* in HSNs. HSN expression of *cca-1* dsRNA potently reverted the suppression conferred by *cca-1(n5209)* (**Figure 6G**). We introduced this RNAi transgene into wild-type HSNs and observed only a modest effect on egg-laying behavior (**Figure 6—figure supplement 2**), that was similar to the effect of a *cca-1* deletion mutation (**Figure 3D**). These data are consistent with our expectation that knockdown of CCA-1 would revert the suppression of the *egl-6(gf)* phenotype by *cca-1(n5209)*. When we expressed *cca-1(RNAi)* in vulval muscles and in VC4 and VC5, we did not observe any significant effect on the suppression of the *egl-6(gf)* phenotype by CCA-1(n5209) channels (**Figure 6H and I**). Together, these data indicate that CCA-1(n5209) mutant T-type channels act in the HSNs themselves. Thus, re-tuning T-type channels renders the cells in which they are expressed insensitive to peptidergic inhibition.

## Re-tuned T-type channels restore activity to serotonin neurons that receive excess peptidergic inhibition

To determine the effects of peptidergic inhibition on the physiology of the egg-laying system, we expressed the high-sensitivity calcium indicator GCaMP6f in the HSNs (**Figure 7A–D**) and the VMs (**Figure 7—figure supplement 1**) of wild-type and mutant animals (**Chen et al., 2013**). We compared the calcium activity of wild-type and *egl-6(gf)* HSNs and found that HSN activity was severely affected by increased peptidergic inhibition. Wild-type HSNs displayed spontaneous calcium transients (**Figure 7A and C**, top), consistent with previous reports (**Shyn et al., 2003**; **Zhang et al., 2008**). By contrast, HSNs of *egl-6(gf)* mutants were largely silent, and only rarely did we observe a calcium transient in these mutant HSNs (**Figure 7A and C**, second panel). Strikingly, the *cca-1 (n5209)* suppressor mutation restored activity to *egl-6(gf)* HSNs (**Figure 7A–D**), but the calcium transients in *egl-6(gf) cca-1(n5209)* mutant HSNs were lower in amplitude and occurred at a higher frequency than those of wild-type HSNs. We observed similar low-amplitude, high-frequency calcium transients in the HSNs of *cca-1(n5209)* single mutants (**Figure 7A and C**).

Because we observed that calcium transients were uncorrelated with egg laying, we hypothesized that the synaptic targets of the HSNs might integrate HSN activity over time. To test this hypothesis, we calculated the cumulative GCaMP signals (cumulative $\triangle F/F$) in HSN somas and axons as a proxy for neuronal activity over time. In HSN somas of *egl-6(gf) cca-1(n5209)* mutants, cumulative $\triangle F/F$ was significantly elevated compared to that measured in the somas of *egl-6(gf)* HSNs but was still lower than cumulative $\Delta F/F$ measured in the somas of wild-type HSNs (**Figure 7B**). In HSN axons,

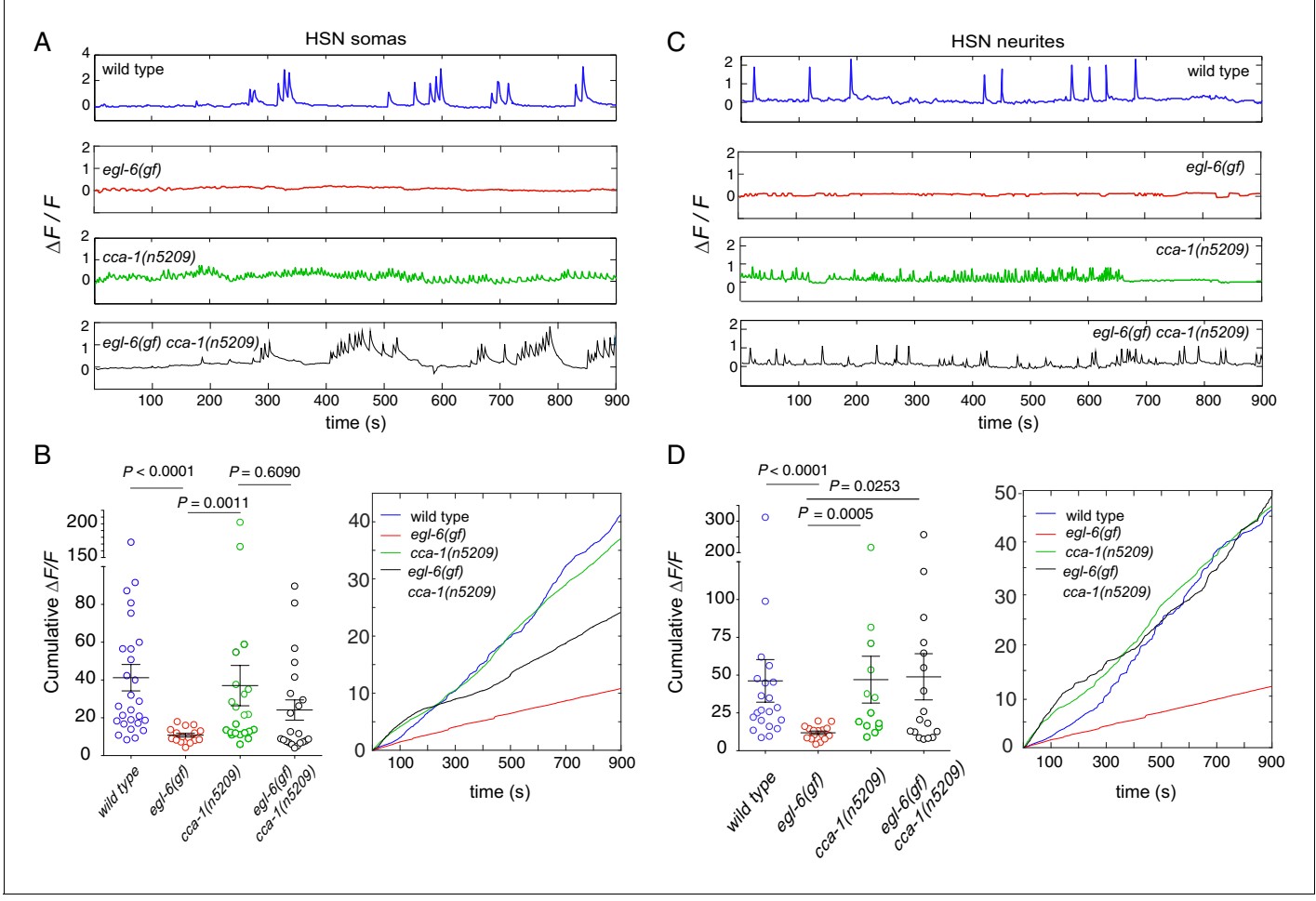

**Figure 7.** Re-tuned channels restore neural activity to inhibited serotonin neurons. (**A**) Representative △F/F traces of spontaneous somatic calcium transients in HSNs of partially restrained adult hermaphrodites. (**B**) Cumulative △F/F of wild-type and mutant HSN somas. Values for individual animals are displayed on the left; mean values for each genotype are displayed on the right. (**C**) Representative △F/F traces of spontaneous neurite calcium transients in the HSNs of partially restrained adult hermaphrodites. (**D**) Cumulative △F/F of wild-type and mutant HSN neurites. Values for individual animals on left; mean values for each genotype on right. 16–22 animals were imaged for all genotypes. Soma: N = 16–27. Neurite: N = 13–22. Calcium imaging of vulval muscles in these strains is shown in *Figure 7—figure supplement 1*.

The following source data and figure supplement are available for figure 7:

**Source code 1.**

**Figure supplement 1.** Excess peptidergic inhibition onto HSNs does not significantly alter spontaneous vulval muscle activity.

however, cumulative △F/F of the *egl-6(gf) cca-1(n5209)* neurons was strikingly similar to that measured in wild-type axons. This complete restoration of cumulative activity in the HSN neurite correlated with the complete restoration of reproductive behavior conferred to *egl-6(gf)* mutants by the *cca-1(n5209)* mutation (*Figure 7D*). These data show that *cca-1(n5209)* restores activity to HSNs that have been silenced by excess peptidergic inhibition. These data further suggest that T-type channels regulate axonal, that is, presynaptic, calcium in serotonergic HSNs and that altering the voltage-gating of T-type channels may have a larger effect on axonal/presynaptic calcium than it does on calcium signaling in the HSN cell body.

## Discussion

A specific and coordinated change in voltage-dependent activation and inactivation of T-type calcium channels restores physiological activity to neurons of the reproductive neuromusculature that have been silenced by peptidergic inhibition. Importantly, these altered channels also restore wild-type reproductive behavior to mutants with excess endogenous peptidergic inhibition. Put another way, changing the voltage dependencies of T-type calcium channels functionally disengages the neurons in which they are expressed from a powerful modulatory input. Furthermore, in the absence of excess peptidergic inhibition, this change in T-type channel regulation causes no effect on reproductive behavior. How might a seemingly simple adjustment of channel properties exert such a powerful and specific effect on a neural circuit?

### Mechanisms by which T-type currents might gate inhibitory peptidergic modulation

We propose two mechanisms by which changing the voltage-dependencies of activation and inactivation could restore wild-type function to serotonin neurons inhibited by neuropeptide modulators.

1) The window current might play a critical role in setting the resting potential of serotonergic HSNs. In the voltage range covered by the overlap of activation and inactivation curves, not all T-type channels fully inactivate, and so some continue to flux ions (*Crunelli et al., 2005*; *Uchino et al., 2013*; *Williams et al., 1997*). The T-type window current is known to contribute to the resting membrane potential of vertebrate thalamocortical and reticular neurons (*Dreyfus et al., 2010*). We previously showed that inhibitory neuropeptides require GIRK-like potassium channels to inhibit HSNs, and it is therefore likely that inhibitory neuropeptides hyperpolarize HSNs (*Emtage et al., 2012*). Re-tuned T-type channels might be active at these hyperpolarized potentials and thereby pass a depolarizing current that brings the membrane potential of HSNs back to their pre-modulator baseline. Importantly, in the absence of inhibitory peptides, the resting potential of HSNs would be such that re-tuned T-type channels are inactivated. Re-tuned T-type channels could have little effect on circuit function in the absence of hyperpolarizing modulatory inputs, which is consistent with our observation that re-tuned T-type channels have little effect on *C. elegans* reproductive behavior in the absence of peptidergic inhibition.

2) Re-tuned T-type channels might boost small depolarizing inputs onto HSNs that are hyperpolarized by peptidergic inhibition and allow those depolarizing inputs to drive behavior. Because T-type channels are low-voltage-activated, they need only a small depolarization to open, and the current they pass can then bring other ion channels to threshold. Such a role for T-type channels in boosting excitatory inputs has been observed in vertebrate neurons (*Llinás and Yarom, 1981*; *Perez-Reyes, 2003*). T-type channels can also boost depolarizations generated by cell-intrinsic mechanisms, for example in cardiac myocytes (*Hagiwara et al., 1988*; *Zhou and Lipsius, 1994*; *Mangoni et al., 2006*) and their *C. elegans* counterparts, pharyngeal muscle cells (*Shtonda and Avery, 2005*). A key distinction between this model and the model proposed above is that here, re-tuned T-type channels do not affect the resting potential of cells that receive hyperpolarizing modulatory input. Rather, the re-tuned channels would boost excitatory inputs that would otherwise be below threshold for triggering behavior. The ability of T-type channels to boost excitation would also be limited to a hyperpolarized range of membrane potentials because of channel inactivation. In the absence of neuropeptide-driven hyperpolarization, re-tuned calcium channels would likely be inactivated and not affect cellular or circuit responses to excitatory inputs. Again, this model is consistent with our observation that re-tuned T-type channels only affect behavior when HSNs receive peptidergic inhibition.

### How T-type channels alter the activity of HSNs suggests an integrative mechanism in targets receiving serotonergic input

Re-tuned T-type channels restored wild-type behavior to animals in which HSNs receive excess inhibition, but the pattern of HSN activity in these animals was strikingly different from that in the wild type. In the wild type, we observed low-frequency, high-amplitude calcium transients in HSN cell bodies and neurites. Under the conditions we used to record HSN activity these calcium transients did not correlate with egg laying or muscle contractions (data not shown), and only rarely did we observe egg laying by immobilized animals. Recently, Collins *et al.* recorded HSN activity in freely

moving animals and showed that HSNs are active during periods of behavioral quiescence (*Collins et al., 2016*). Notably, our data and those of Collins *et al.* are inconsistent with a model in which the HSNs function as typical motor neurons to acutely drive target muscle contractions.

Peptidergic inhibition through the neuropeptide receptor EGL-6 strongly affected calcium transients in HSNs; almost none of the *egl-6(gf)* mutant HSNs displayed any calcium transients. On its own, the *cca-1* suppressor mutation strikingly reduced the amplitude and increased the frequency of HSN calcium transients. These high-frequency, low-amplitude transients in *cca-1* mutant HSNs were not affected by increased peptidergic inhibition. That is, HSNs carrying re-tuned T-type channels were functionally disengaged from inhibitory peptidergic modulation at the level of cell physiology as well as behavior.

Although the calcium transients of wild-type and *cca-1* mutant HSNs were clearly different, we found that the cumulative activity of *cca-1* mutant HSNs was comparable to that of wild-type HSNs. These data suggest that reproductive behavior is not driven by fast, phasic signaling from HSNs, as might be suggested by the direct innervation of vulval muscles by serotonergic HSNs. Instead, our data indicate that HSN targets integrate signals from HSNs over longer periods. On average, bursts of reproductive behavior are separated by twenty to thirty minutes of quiescence, suggesting that the timescale over which such an integrative mechanism would function is on the order of tens of minutes. This model raises the interesting question of what molecular and biophysical mechanisms allow the reproductive circuit to integrate HSN activity during behavioral quiescence.

## T-type calcium channels might play a widespread role in gating the response to modulatory inputs to neural circuits

Unlike *C. elegans*, whose genome encodes only one T-type channel, vertebrates possess three genes that encode T-type channels, each of which is expressed in many central circuits. The duplication of T-type calcium genes in vertebrate phyla could reflect expanded and diversified roles for T-type channels that are analogous to their function in the *C. elegans* serotonin system. If so, different T-type channel isoforms might be involved in the gating of multiple modulatory systems in the vertebrate brain. Genetic studies of human populations have found associations between T-type channel variants and neurological or psychiatric disorders (*Cheong and Shin, 2013*). Alleles of T-type channel genes are strongly linked to heritable epilepsy (*Kim et al., 2001*; *Heron et al., 2004*; *Khosravani et al., 2005*; *Singh et al., 2007*; *Powell et al., 2009*) and neuropathic pain (*Bourinet et al., 2016*). A T-type channel variant is also associated with autism spectrum disorders (*Splawski et al., 2006*). Targeted genetic studies of T-type channels in the mouse brain confirm that these channels play roles in diverse brain circuits. T-type channels are required in thalamic circuits for normal sleep, in midbrain centers for central processing of nociception, and in cerebellar circuits for motor learning (*Anderson et al., 2005*; *Park et al., 2010*; *Ly et al., 2013*). Our study raises the intriguing possibility that in these brain regions altering T-type channels might affect receptivity of neural circuits to modulatory inputs and that defective gating of neuromodulation might be an underlying cause of pathologies associated with T-type channel dysfunction.

T-type channel isoforms are differentially regulated by diverse intracellular signaling systems, including systems that couple to protein kinases C and A and heterotrimeric G proteins (*Wolfe et al., 2003*; *Park et al., 2003*, *2006*; *Kim et al., 2006*; *DePuy et al., 2006*; *Hu et al., 2009*; *Perez-Reyes, 2010*). Furthermore, T-type channel genes encode transcripts that display a high degree of alternative splicing to generate multiple channel isoforms from a single locus (*Senatore et al., 2014*). Neurons can, therefore, remodel the complement of T-type channels either through genetic programs and mechanisms that couple neural activity to mRNA splicing, or acutely regulate T-type channels through canonical second messenger signaling pathways and effectors (*Murbartián et al., 2004*; *Senatore et al., 2014*). Because of this regulation, in the vertebrate brain a gating function of T-type channels with respect to neuromodulation would be subject to a high degree of selective control and possess a large capacity for plasticity.

The hypothesis that T-type channels gate neuromodulatory inputs to circuits implies that T-type calcium channels could be important therapeutic targets for psychiatric disorders linked to defects in neuromodulatory systems. Compounds that directly target monoamine and neuropeptide signaling systems are in widespread use for the treatment of psychiatric disease. Based on our studies of a model serotonergic circuit in *C. elegans,* we suggest that compounds that alter the voltage-dependencies of T-type channels might synergize with therapeutics that directly target monoamine

signaling in the brain. Furthermore, our studies demonstrate the possibility that globally manipulating T-type channel function in the brain could affect specific circuits. T-type channels are widely expressed in the *C. elegans* nervous system, as they are in the vertebrate brain, but in *C. elegans*, re-tuning T-type channels by mutation had a striking effect on the modulation of a circuit that controls reproductive behavior. Our data suggest that the origins of this apparent specificity are in the combination of in the biophysical basis of inhibition of serotonin neurons and the manner in which serotonin signals are processed by a downstream circuit. It is highly likely that similar cellular and circuit mechanisms exist in the vertebrate brain, and when the pharmacological tools become available it will be important to test whether manipulating vertebrate T-type channels restores function to neural circuits in and behavior of animal models of psychiatric disorders associated with monoamine dysfunction.

## Materials and methods

### Nematode culture, strains, and maintenance

*C. elegans* hermaphrodites were grown on nematode growth medium (NGM) agar at 20°C on *Escherichia coli* OP50 as described previously (*Brenner, 1974*). Wild-type *C. elegans* were the Bristol N2 strain. *n5209* heterozygotes were generated by crossing FQ67 *egl-6(n592) mec-7(e1506) lin-15AB(n765)* with FQ46 *egl-6(n592) cca-1(n5209)* and identified as non-Muv cross-progeny. A full list of strains can be found in *Supplementary file 3*.

### Screening for suppressors of the egg-laying defect caused by increased peptidergic inhibition

The strain MT1222, which carries the *egl-6(n592gf)* mutation, was mutagenized with EMS as described (Brenner, 1974). *egl-6(gf)* mutants are bloated with unlaid eggs. Progeny in the second generation (F2) after mutagenesis were screened for alleviation of this bloating, strains were established from non-bloated F2 progeny and those strains were subsequently scored for an egg-retention phenotype and for the developmental stage of newly laid eggs as described below. Approximately 50,000 haploid genomes (more than 100,000 F2 progeny) were screened, yielding six strains with restored egg-laying behavior.

### Fine mapping of the suppressor mutation *n5209*

Initial outcrosses indicated that the *n5209* suppressor mutation was linked to the *egl-6* locus on chromosome X. We confirmed that *n5209* is on chromosome X by measuring its position relative to the interval defined by the genes *mec-7* and *lin-15* using conventional three-factor crosses (Brenner, 1974). These studies showed that *n5209* was close to or right of *mec-7*. Next, we used whole genome sequencing (WGS) to determine the sequence of *egl-6(gf) n5209* genomic DNA and identified sequence variants that were present in suppressed mutants but not in the wild-type or in the parental strain MT1222. We designed PCR primers that detect these sequence variants as previously described (*Drenkard et al., 2000*), and we used these variants as markers to map *n5209* after outcross of *egl-6(gf) n5209* mutants to either N2 or MT1222 and recovery of egg-laying defective (Egl non-Sup) or suppressed egg-laying defective (Egl Sup) recombinants.

### Analysis of whole-genome sequencing data

To identify the *cca-1(n6122)* mutation, sequence variants were identified using the Genome Analysis ToolKit (*McKenna et al., 2010a*) according to best practices described in (*DePristo et al., 2011a*; *Van der Auwera, 2013*). Variant filtering to identify sequence variants in *n6122* mutants was performed using snpEff and snpSift (*Cingolani, 2012*).

### Genotyping mutant *C. elegans*

Alleles were verified via PCR, using primers that specifically amplified sequences containing point mutations (*Drenkard et al., 2000*). A full list of genotyping primers can found in *Supplementary file 4*.

## Generation of transgenes and plasmids

A *cca-1* fosmid reporter was generated by inserting GFP into the first exon of *cca-1*. GFP carrying an FRT-flanked GalK selection cassette in a synthetic intron was amplified from the plasmid pBALU1 using the following primers:

KZ259 5'-GGAAACTGAACTTACCGCTTTCATTCAACC-3'
KZ262 5'-GGACTGAAAACTGCGCAGTTCCGGAACTGGCTGACGTAGCA-3'

Fosmid recombineering was performed as previously described (*Tursun et al., 2009*). A 25.8 kb amplicon generated from the recombineered fosmid was used for transgenesis. Clones of mammalian Cav3.1 in plasmid pGEMHEA were a generous gift from Jung-Ha Lee (Sogang University, Korea). The *n5209* mutation was introduced into mammalian Ca$_v$3.1 to generate pKEZ18 using the Quik-Change II XL Site-Directed Mutagenesis Kit (Stratagene) and the following primers:

KZ28 5'-CATTACTGCTGGACACCTTGTCTATGCTGGGCAACGTCCT-3'
KZ29 5'-CAGGACGTTGCCCAGCATAGACAAGGTGTCCAGCAGTAATG-3'

Other plasmids for *C. elegans* transgenesis were made using conventional methods. A full list of constructed plasmids can be found in *Supplementary file 4*. Sequences are available in *Supplementary file 2*.

## Generation of transgenic *C. elegans* strains

Transgenic animals were generated via microinjection as previously described (*Mello et al., 1991*). A full list of injected constructs and their concentrations can be found in *Supplementary file 4*.

RNAi lines were made with the following constructs:

HSN-specific *cca-1* knockdown: pKEZ67 and pKEZ93, which were made using the *egl-6a::irk-1 RNAi* construct from *Emtage et al. (2012)* as a template.

VC-specific *cca-1* knockdown: pKEZ63 and pKEZ64

VM-specific *cca-1* knockdown: pKEZ65 and pKEZ66

## Behavior assays

*C. elegans* hermaphrodites aged 24–30 hr post-L4 were used for behavior assays. Egg-stage assays were performed as follows: Animals were placed on a lawn of OP50 and allowed to lay eggs for 1 hr at 20°C. Mothers were removed, and embryos were scored using a high power dissecting scope (Leica M165FC). Embryos were assigned to one of seven categories: 1–8 cell, 9–25 cell, >26 cells but not yet undergone morphogenesis, comma stage, two-fold, three-fold, and embryos that hatched during the observation period and appeared as L1 larvae.

Egg-laying data was collected on at least three separate days to an *N* of at least 50 embryos. All behavior was done with paired wild-type controls.

For video tracking, animals were recorded while on a 1 cm lawn of OP50 for 2 hr using a one megapixel CCD camera (UniBrain; San Ramon, CA) on a stereomicroscope (Motic; Hicksville, NY). Videos were analyzed offline by manually counting egg-laying events. Animals that left the field of view during video tracking were excluded from analysis. We fit the data to the model described in *Waggoner et al. (1998)* in MatLab with the following equation using the Nonlinear Least Squares method: 1-(((p*(a-b)/(a-p*b))*exp(−1*a*x))+((a*(1 p)/(a-p*b))*exp(−1*p*b*x))).

## Microscopy

Young adults were anesthetized with 30 mM sodium azide and mounted on 2% agarose pads made in M9 medium. Z-stacks were obtained with either a Zeiss LSM510 or LSM700 confocal microscope, and maximum-projection images were created using ImageJ (W. S. Rasband, ImageJ, National Institutes of Health, Bethesda, MD; http://imagej.nih.gov/ij/; 1997–2011).

## Xenopus laevis *oocyte electrophysiology*

Capped mRNA was made using the mMessage mMachine T7 Kit (Ambion; Austin, TX). *X. laevis* oocytes were injected with 60–80 ng sense Cav3.1 or Cav3.1(n5209) cRNA using a Drummond Nano-ject pipette injector (Parkway, PA) attached to a Narishige micromanipulator (Tokyo, Japan). Injected oocytes were incubated at 17°C for 3–5 days before recording. Oocytes were kept in ND96 (96 mM NaCl 2.5 mM KCl 1 mM MgCl$_2$-6H$_2$O 5 mM HEPES 1.8 mM CaCl$_2$-2H$_2$O pH 7.6 adjusted with NaOH) with 2.5 mM sodium pyruvate and 1:100 penicillin-streptomycin. Whole-cell current

recordings were made using the two-electrode voltage-clamp method (Warner Instruments; Hamden, CT). Electrodes had a resistance between 0.2 and 1.0 MΩ. Oocytes were continuously superfused with Barium Hydroxide (10 mM Ba(OH)$_2$1 mM KOH 90 mM NaOH 5 mM HEPES pH 7.4 adjusted with methane sulfonic acid). Data were acquired with a Digidata 1440 digitizer (Molecular Devices; Sunnyvale, CA) and analyzed off-line with Clampfit (Molecular Devices) and MATLAB (Simulink; Natick, MA). Each oocyte was only included once in the data analysis, that is, there are no technical replicates. In order to be included in the data analysis, oocytes had to exhibit a leak current of less than −0.1 µA when clamped at −90 mV, and they had recover to normal resting membrane potential after the activation and inactivation protocols were done. V$_{50}$ for activation and inactivation and the slope factor, $k$, as well as the confidence intervals, were computed using the Boltzmann sigmoidal fit tool in GraphPad Prism (La Jolla, CA). All experiments using *Xenopus laevis* were performed according to guidelines of the Committee on Animal Care at New York University Langone Medical Center.

## Calcium imaging

Adult animals with a single row of eggs in the uterus were selected for imaging. *egl-6(gf)* animals were imaged 18–24 hr post larval-stage 4 (L4). Wild-type, *cca-1(n5209)*, and *egl-6(gf) cca-1(n5209)* animals were imaged 24–30 hr post-L4. Animals were mounted on 5% agarose pads made in M9 medium, immobilized using Vetbond veterinary surgical glue (3 M; St. Paul, MN), and placed under a coverslip. All videos were taken using a 10x objective on a Zeiss Axioscope Imager.M2 using Micromanager (MMstudioversion1.4.18). GCaMP6f was excited by 473 nm light using an EXFO metal halide epifluorescence light source at 12% light intensity. Images were acquired with a cooled CCD camera (Andor; Belfast, Northern Ireland). Animals were imaged at 0.5 Hz for 15 min. Only animals that survived the imaging protocol were included in the study. At least 16 animals per genotype were analyzed. If both HSNs of one animal could be clearly seen, then both were included in the analysis. Neurites were not always visible, even when the soma was. Each animal was only imaged once. GCaMP6f fluorescence was analyzed as follows: Mean background fluorescence was subtracted from mean fluorescence in the region of interest (ROI). Baseline fluorescence in the ROI was determined by taking a linear regression of the entire video and subtracting it from the raw ROI data. This baseline was used to adjust fluorescence signals within the ROI for photobleaching during the experiment. To calculate the cumulative $\triangle F/F$ of each trace, the absolute value of the difference between ROI fluorescence at each time point was summed. The MatLab scripts we used to analyze the data as described can be found in *Figure 7—source code 1*.

## Statistical analysis

All *P*-values for egg-laying behavior are listed in *Supplementary file 1*. For egg-laying behavior and for calcium imaging, the Wilcoxon Mann-Whitney Rank Sum Test was used because the data were not assumed to have a normal distribution.

## Acknowledgements

*n5209* and *n6122* mutants were isolated in the laboratory of Bob Horvitz, MIT Department of Biology, and the authors thank him for supporting early stages of this project. The authors also thank Sonya Aziz-Zaman for assistance with mapping and cloning *cca-1(n5209)*; Jin-Yong Park for sharing expertise in recording T-type currents; Jung-Ha Lee for providing the Ca$_v$3.1 expression construct; Yuji Kohara for *C. elegans* cDNA clones; Katherine Nagel and David Schoppik for discussions and help with data analysis; Kathryn Allaway for assistance with calcium imaging, and all members of the Ringstad laboratory for helpful comments. Some strains were provided by the CGC, which is funded by NIH Office of Research Infrastructure Programs (P40 OD010440). This work was supported by NIGMS R01–098320 and NIGMS R01–113182 to NR, and NINDS F31NS089232 to KEZ, who also received support from NIMH training grant T32MH096331.

# Additional information

## Funding

| Funder | Grant reference number | Author |
|---|---|---|
| National Institute of General Medical Sciences | R01-098320 | Niels Ringstad |
| National Institute of General Medical Sciences | R01-113182 | Niels Ringstad |
| National Institute of Neurological Disorders and Stroke | F31NS089232 | Kara E Zang |

The funders had no role in study design, data collection and interpretation, or the decision to submit the work for publication.

## Author contributions

KEZ, Conceptualization, Data curation, Software, Formal analysis, Investigation, Methodology, Writing—original draft, Writing—review and editing; EH, Data curation, Formal analysis, Investigation, Methodology; NR, Conceptualization, Resources, Data curation, Software, Formal analysis, Supervision, Funding acquisition, Validation, Investigation, Visualization, Methodology, Writing—original draft, Project administration, Writing—review and editing

## Author ORCIDs

Niels Ringstad, http://orcid.org/0000-0002-8679-2269

## Ethics

Animal experimentation: Animal subjects (Xenopus laevis frogs) were used in strict accordance with the recommendations in the Guide for the Care and Use of Laboratory Animals of the National Institutes of Health. All animals were handled according to the approved institutional animal care and use committee (IACUC) protocol #131102-03.

# Additional files

## Supplementary files

• Supplementary file 1. List of all *p*-values.

• Supplementary file 2. Plasmid sequences.

• Supplementary file 3. List of strains used in this study.

• Supplementary file 4. List of primers and plasmids used in this study.

• Supplementary file 5. Sequence of RNAi seed.

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
