## [Decision Letter]

Thank you for submitting your article "Inhibitory peptidergic modulation of *C. elegans* serotonin neurons is gated by T-type calcium channels" for consideration by *eLife*. Your article has been reviewed by two peer reviewers (one has agreed to reveal his identity, Michael Koelle), and the evaluation has been overseen by Oliver Hobert as the Reviewing Editor and Richard Aldrich as the Senior Editor.

The reviewers have discussed the reviews with one another and the Reviewing Editor has drafted this decision to help you prepare a revised submission.

This is a very fine paper that utilizes genetics, electrophysiology, and Ca^2+^ imaging in the nervous system of intact animals to investigate how neuropeptides modulate activity of specific serotonergic neurons in the *C. elegans* egg-laying circuit. This neural circuit is very highly developed as an experimental system to study how specific genes and signals regulate the activity of the circuit and the behavior it controls. The authors take advantage of all the available tricks in this system to carry out studies that would be difficult or impossible outside of *C. elegans*, or even within *C. elegans* in less well-studied neural circuits. The state-of-the-art nature of this study within the neural circuit field in itself makes this study of considerable interest. The results of the study make it of further interest. These results arise from an unbiased genetic screen for genes required for response to neuropeptide inhibition of neural activity, and therefore we find them to be of particular interest and significance, since this approach often leads to novel and physiologically significant findings. The results are indeed novel and interesting and lead the authors to raise the hypothesis that T type calcium channels generally are specific and key regulators of the responsiveness of neurons to neuromodulators.

Below is a list of comments and technical concerns that the authors should address. Only one of them (item 1) requires additional experimental work (estimated 2-3 weeks). The remaining items on the list can be addressed with minor alterations to the writing.

1) There is a technical concern about Figure 6. Although it is a bit of a puzzle to figure this out from the information supplied in the supplemental Materials and methods, it appears that the plasmids injected to carry out cell-specific RNAi knockdown of *cca-1* in the HSNs are pKEZ93 and pKEZ67, which express antisense and sense versions of *cca-1* from the fairly HSN-specific *egl-6a* promoter. The effect of injecting these constructs is to make the animals go from relatively wild-type egg laying to becoming egg-laying defective, and the authors interpret this as showing the knockdown of *cca-1(n5209)* in the HSN causes this effect, i.e. *cca-1(n5209)* is necessary in the HSN to suppress *egl-69(gf)*. Unfortunately, there is possibly a more mundane explanation. A number of labs working on the HSN neurons have found that high copy transgenes carrying HSN-expressed promoters (including the *egl-6* promoter) can interfere with HSN development. Development of the HSN seems to be unusually susceptible to disruption, so the types of manipulations that would cause no worries for other neurons are indeed potentially problematic for the HSN. Disrupting HSN development would lead to the egg-laying defective phenotype, as the authors themselves show in Figure 5. So, I feel that interpretation of Figure 6 would be *strongly* enhanced by a control showing that "*cca-1(RNAi)* in the HSNs" by itself does not cause an egg laying defective phenotype. I would like to see the authors coinject pKEZ93 and pKEZ67 into a background wild-type for both *egl-6* and *cca-1*, at the same concentrations used in Figure 6, do their staging of freshly laid egg assay on several lines generated in this way, and determine if the RNAi transgene itself causes an egg-laying defective phenotype. This is a relatively quick experiment to do – I would estimate 2-3 weeks.

2) The genetic screen that produced *n5209* is described only in passing, and no technical details are given, and it is cited as a personal communication. This is work done by the last author, Ringstad, as a postdoc in the Horvitz lab. This is a bit of an unusual way to handle this issue. Isolation of the mutant is the entire basis of the paper, so it is an odd thing to leave out. There needs to be more precise methodology for this screen and some details of the results available to other workers who might want to use this knowledge to design other suppressor of Egl screens.

3) Figure 1 panels F and G show a very nice analysis. However, I have two comments. First, the authors conclude that the intracluster time constant is significantly changed, while the intercluster time constant is not. Is there an appropriate statistical analysis that could be used to determine if the data actually show this? Second, I would like to comment that it appears from Figure 1 (although no quantitative data are shown on this point) that *egl-6(gf)* worms carry many more unlaid eggs than do *egl-6(gf) n5209* worms. I wonder if the apparent increase in the intracluster time constant in the latter worms is simply a byproduct of the fact that there are less eggs to lay, so that egg-laying muscle contractions are less likely to productively release eggs. That would be a different interpretation from thinking that *n5209* has a specific effect on regulation of intracluster intervals. The authors, to their credit, do *not* try to conclude that *n5209* has a specific effect on regulation of intracluster intervals, but rather simply conclude that the overall pattern of egg laying (alternating inactive and active phases) remains intact.

4) Subsection “*n5209* affects the T-type calcium channel CCA-1”, first paragraph. If the authors used any methods for SNP mapping or analysis of whole genome sequencing based on published methods developed by others, it would be nice if they cited that work in the Results or methods sections. I did eventually find such citations in the Figure 2—figure supplement I legend, but they were hard to find.

5) Subsection “*n5209* affects the T-type calcium channel CCA-1”, first sentence. This sentence in the Results says there was only one protein coding sequence variant in the interval to which *n5209* was mapped. Figure 2 and its legend shows that *ubc-17*, C36B7.5, and F49E2.1 are in an interval to which *n5209* mapped and all three contained coding sequence variants that by methods not explained were shown not to affect the *egl-6(gf)* phenotype and thus to not be *n5209*. The apparent discrepancy between the language in the Results and in the figure/figure legend should be resolved so that this issue does not confuse the readers. Perhaps the resolution has to do with *cca-1* being the only gene with sequence variants in both sequenced *egl-6(gf)* suppressor alleles, but that doesn't become apparent until one has read on to later in the paragraph.

6) The argument that the point mutation in *cca-1* is the causative mutation that suppresses *egl-6(gf)* appears to depend principally on this mutation having been isolated twice in two independently isolated suppressor strains. This is a reasonable argument, but there are potential holes in it (i.e. both mutations arose from a single "jackpot" mutation and aren't really independent). I'd prefer it if the authors explicitly address the issue of the criteria they used to determine that the *cca-1* point mutation is indeed the *egl-6(gf)* suppressor mutation.

7) Figure 3 legend. The authors write that one copy of *n5209* was sufficient to suppress *egl-6(gf)*. I suggest they change the wording to say it was sufficient to *partially* suppress *egl-6(gf)*. Figure 1 shows that two copies of *n5209* appear to fully suppress, and one copy does not cause nearly as much suppression. When I read the unmodified "suppress", I feel that full suppression is implied, and that simply saying "suppress" is not accurate here, especially since the authors emphasize the weaker "partial suppression" shown in Figure 3.

8) Subsection “*n5209* affects the T-type calcium channel CCA-1”, last sentence. "*n5209* does not cause loss of *cca-1* function". I feel this concluding statement is an oversimplification of the results. The fact is that the deletion mutation does weakly partially suppress *egl-6(gf)*, and *n5209*/+ and *n5209/n5209* cause stronger suppression of *egl-6(gf)*. So, the *n5209* phenotype shares some similarity with the null phenotype and *n5209* may have some loss of function character. I'd be fine with saying "*n5209* does not *simply* cause loss of *cca-1* function."

9) In the subsection “*n5209* alters the voltage dependency of the voltage-gated T-type calcium channel”, the authors put a P to S point mutation into a mammalian ortholog of *cca-1* analogous to the *n5209* P to S mutation in *cca-1*, and characterize effects of the mutation on mammalian channel activity. It is implied, but not stated explicitly, that the authors either didn't try or tried and failed to get *C. elegans* CCA-1 functionally expressed in *Xenopus oocytes*, decided to functionally characterize the mammalian channel and its mutant instead, and that they are inferring that the P to S point mutation has similar effects on the worm and mammalian channels. There is nothing wrong with the authors' reasoning, but instead of making the readers read between the lines and infer all this, I would prefer it if the authors simply said these things explicitly. Perhaps this is a subjective matter of style, but I prefer things said out in the open so that reading the paper is less of a puzzle for the reader to solve.

10) A separate technical concern on Figure 6: The authors show a N value for the number of eggs staged. They do not indicate (as far as I can find) whether all the eggs assayed in each Figure panel came from a single transgenic line, or from multiple lines. Because there is line to line variation in expression levels of *C. elegans* extrachromosomal transgenes, it would be better if the authors analyzed multiple lines, and then either pool the data from several lines, tell us they are showing data from a representative line, or if they are picking their "best" line, tell us that. In any case, some more explicit information on how the experiment was done is in order.

11) The Discussion is largely speculative, although speculation is appropriate and welcome in a Discussion. The first section (subsection “Mechanisms by which T-type currents might gate inhibitory peptidergic modulation”) speculates about how retuned T type channels affect the HSN, but because the HSN is not so far accessible to direct electrophysiological recording, the ideas explored remain untestable. The second section (subsection “How T-type channels alter the activity of HSNs suggests an integrative mechanism in targets receiving serotonergic input”) analyzes results of the calcium imaging. Because the calcium imaging was done with immobilized animals on agarose pads, a condition in which "normal" egg laying behavior with active and inactive phases cannot occur and in which HSN activity is not correlated with egg laying behavior, there are limits to how well one can interpret what the data mean for normal egg laying behavior. The final Discussion section hypothesizes that T type Ca^2+^ channels generally "gate" neuromodulation and therefore could be good drug targets for psychiatric diseases that seem to involve neuromodulation. This is an interesting idea, but I am not clear how one would achieve a very specific effect with such drugs since T type channels are presumably expressed very widely in the brain.

12) In Figure 1, it was not clear how the timing parameters for egg-laying were determined (the Methods says using MATLAB scripts, but it is not clear how these work). Also, I'm not sure the plot in Figure 1 illustrates the phenotype of the mutant very well; a histogram of interval times on a log scale would probably work better.

13) I think it would have been useful for the authors to try to relate the behavioral phenotype to the neural phenotype seen in HSN calcium imaging. For instance, recent papers have linked individual egg-laying events to large calcium spikes in HSN, so perhaps the fact that the *cca-1* mutations cause high frequency but low amplitude calcium activity in HSN is consistent with the lower rate of egg-laying within clusters that they observe behaviorally.

14) In the calcium imaging experiments (Figure 7), were any egg-laying events observed? If so it would be interesting to see how they correlate with the calcium traces; if none were seen this should be noted.

---

## [Author Response]

*[…] Below is a list of comments and technical concerns that the authors should address. Only one of them (item 1) requires additional experimental work (estimated 2-3 weeks). The remaining items on the list can be addressed with minor alterations to the writing.*

*1) There is a technical concern about Figure 6. Although it is a bit of a puzzle to figure this out from the information supplied in the supplemental Materials and methods, it appears that the plasmids injected to carry out cell-specific RNAi knockdown of cca-1 in the HSNs are pKEZ93 and pKEZ67, which express antisense and sense versions of cca-1 from the fairly HSN-specific egl-6a promoter. The effect of injecting these constructs is to make the animals go from relatively wild-type egg laying to becoming egg-laying defective, and the authors interpret this as showing the knockdown of cca-1(n5209) in the HSN causes this effect, i.e. cca-1(n5209) is necessary in the HSN to suppress egl-69(gf). Unfortunately, there is possibly a more mundane explanation. A number of labs working on the HSN neurons have found that high copy transgenes carrying HSN-expressed promoters (including the egl-6 promoter) can interfere with HSN development. Development of the HSN seems to be unusually susceptible to disruption, so the types of manipulations that would cause no worries for other neurons are indeed potentially problematic for the HSN. Disrupting HSN development would lead to the egg-laying defective phenotype, as the authors themselves show in Figure 5. So, I feel that interpretation of Figure 6 would be strongly enhanced by a control showing that "cca-1(RNAi) in the HSNs" by itself does not cause an egg laying defective phenotype. I would like to see the authors coinject pKEZ93 and pKEZ67 into a background wild-type for both egl-6 and cca-1, at the same concentrations used in Figure 6, do their staging of freshly laid egg assay on several lines generated in this way, and determine if the RNAi transgene itself causes an egg-laying defective phenotype. This is a relatively quick experiment to do – I would estimate 2-3 weeks.*

We agree that these are important issues. First, we apologize for the lack of clarity with respect to what reagent was used to perform cell-specific RNAi. We have amended the manuscript to make clear that we are using an *egl-6* promoter that we previously used to knock down the potassium channel gene *irk-1* (Emtage et al.2012), and we have included in the Methods section more detail about the RNAi constructs. We have included details in the main methods (subsection “Generation of transgenic *C. elegans* strains”), the legend to Figure 6 (subsection “T-type calcium channels regulate function of the serotonergic neurons that drive female reproductive behavior”), and [Supplementary-material SD13-data]

We note that our published study shows that this *egl-6* promoter driving *irk-1* RNAi causes suppression of egg-laying defects, which strongly indicates that the promoter itself does not have a strong deleterious effect on egg-laying behavior.

Second, to directly address the reviewer’s comment, we have collected additional data. We had generated transgenic animals in which this identical promoter drives expression of fluorophores, not RNAi seeds. We measured egg-laying behavior of these transgenics and found that they have near- normal egg-laying behavior (see Figure 8). Also, we have introduced the *Promegl-6a::cca-1(RNAi)* transgene used to test *cca-1* site-of-action into otherwise wild-type animals. As the reviewers suggested, we tested whether this transgene has strong effects on egg laying by the wild type. We found that RNAi of *cca-1* only has a mild effect on egg laying by the wild-type whereas in *egl-6(gf) cca-1(n5209)* mutants RNAi of *cca-1* causes a strong Egl phenotype *i.e.* reverts the suppression caused by *n5209*. These data strongly support our interpretation of the RNAi data in the manuscript. We note that *cca-1* RNAi does have a small effect in the wild type. This is consistent with our observation that *cca-1* loss-of-function mutants display a weak egg-laying defect (Figure 3), and we have made note of this is the manuscript.

Author response image 1.**DOI:**
http://dx.doi.org/10.7554/eLife.22771.029

These data indicate that the *egl-6a* promoter used for these studies does not affect the function of the egg-laying system on its own and supports the conclusion that the effects we see are attributed to knock-down of *cca-1* in HSNs. We have added these data to the manuscript as a supplement to Figure 6 (Figure 6—figure supplement 2).

*2) The genetic screen that produced n5209 is described only in passing, and no technical details are given, and it is cited as a personal communication. This is work done by the last author, Ringstad, as a postdoc in the Horvitz lab. This is a bit of an unusual way to handle this issue. Isolation of the mutant is the entire basis of the paper, so it is an odd thing to leave out. There needs to be more precise methodology for this screen and some details of the results available to other workers who might want to use this knowledge to design other suppressor of Egl screens.*

We apologize for the cursory description of the *egl-6* suppressor screen, which was indeed lacking sufficient details. We have added a description of the screen to the Methods section. We have also removed the citation of the screen as a personal communication, as per the reviewers’ request.

*3) Figure 1 panels F and G show a very nice analysis. However, I have two comments. First, the authors conclude that the intracluster time constant is significantly changed, while the intercluster time constant is not. Is there an appropriate statistical analysis that could be used to determine if the data actually show this?*

We have now included 95% confidence intervals in Figure 1.

*Second, I would like to comment that it appears from Figure 1 (although no quantitative data are shown on this point) that egl-6(gf) worms carry many more unlaid eggs than do egl-6(gf) n5209 worms. I wonder if the apparent increase in the intracluster time constant in the latter worms is simply a byproduct of the fact that there are less eggs to lay, so that egg-laying muscle contractions are less likely to productively release eggs. That would be a different interpretation from thinking that n5209 has a specific effect on regulation of intracluster intervals. The authors, to their credit, do not try to conclude that n5209 has a specific effect on regulation of intracluster intervals, but rather simply conclude that the overall pattern of egg laying (alternating inactive and active phases) remains intact.*

The reviewers also commented that perhaps the number of eggs carried in uterocan explain the difference in the intracluster time constant between the wild type and *n5209* mutants. Because we have been principally using developmental stage of embryos as a measure of time spent in utero, we had not measured the number of eggs carried by *n5209* mutants. Indeed, we find that *n5209* mutants have fewer eggs in uterothan do wild-type hermaphrodites. Under the conditions that we study behavior, *n5209* mutants carry 7.3 ± 2.3 embryos in utero and wild-type animals carry 11.6 ± 3.4. The reviewer’s explanation is, therefore, plausible. We have included these numbers in the Results section (subsection “n5209 does not change the temporal structure of female reproductive behavior”) and noted that a reduction in the number of eggs in uteromight explain the altered rate constant.

*4) Subsection “n5209 affects the T-type calcium channel CCA-1”, first paragraph. If the authors used any methods for SNP mapping or analysis of whole genome sequencing based on published methods developed by others, it would be nice if they cited that work in the Results or methods sections. I did eventually find such citations in the Figure 2—figure supplement 1 legend, but they were hard to find.*

We agree that the methods for SNP mapping and WGS were difficult to find and apologize for this oversight. We have clarified our description of how the *n5209* mutation was mapped and moved this text to the main methods (subsection “Fine mapping of the suppressor mutation *n5209”*).

*5) Subsection “n5209 affects the T-type calcium channel CCA-1”, first sentence. This sentence in the Results says there was only one protein coding sequence variant in the interval to which n5209 was mapped. Figure 2 and its legend shows that ubc-17, C36B7.5, and F49E2.1 are in an interval to which n5209 mapped and all three contained coding sequence variants that by methods not explained were shown not to affect the egl-6(gf) phenotype and thus to not be n5209. The apparent discrepancy between the language in the Results and in the figure/figure legend should be resolved so that this issue does not confuse the readers. Perhaps the resolution has to do with cca-1 being the only gene with sequence variants in both sequenced egl-6(gf) suppressor alleles, but that doesn't become apparent until one has read on to later in the paragraph.*

In Figure 2, we show three coding sequence variants in addition to the *cca-1* variant, but in the text we say that there was only one coding variant in the interval to which *cca-1* was mapped. These other mutations were used as markers for mapping and could be separated from *n5209* by recombination. We apologize for this confusion, which arises from our poor description of our mapping strategy. We have clarified the text (subsection “*n5209* affects the T-type calcium channel CCA-1”, first paragraph) and the figure legend.

*6) The argument that the point mutation in cca-1 is the causative mutation that suppresses egl-6(gf) appears to depend principally on this mutation having been isolated twice in two independently isolated suppressor strains. This is a reasonable argument, but there are potential holes in it (i.e. both mutations arose from a single "jackpot" mutation and aren't really independent). I'd prefer it if the authors explicitly address the issue of the criteria they used to determine that the cca-1 point mutation is indeed the egl-6(gf) suppressor mutation.*

We agree that the manuscript would benefit from a concise and clear statement of the case in favor of *cca-1* as the affected gene. We have added a synopsis of the evidence that leads us to conclude that *cca-1(n5209)* is the causative mutation. In brief:

A) *cca-1* is the only mutation affecting protein-coding sequence within the interval to which the suppressor maps. We have included in the text recombination frequencies from fine-mapping experiments of *n5209* that show how we defined this genetic interval (subsection “*n5209* affects the T-type calcium channel CCA-1”, first paragraph).

B) *n5209* and *n6122* mutations are both X-linked suppressors of the *egl-6(gf)* phenotype and the corresponding mutants contain identical point mutations in *cca-1*. The reviewers suggest that this might be a “jackpot” mutation, but we are confident this is not the case. First, we know that the two strains come from different P0 plates and are not siblings. Second, we have sequenced the genome of each mutant and find that each strain contains a large number of new mutations, including in loci linked to *cca-1*. These data are shown in Figure 2—figure supplement 1. This is consistent with the hypothesis that *n6122* and *n5209* mutations were the result of independent mutagenic events. We have clarified the text to better explain our data and our conclusion that these mutations are independent (-subsection “*n5209* affects the T-type calcium channel CCA-1”).

C) Finally, our genetic analysis suggests that loss of *cca-1* function would revert the suppression caused by *cca-1(n5209).* RNAi of *cca-1* shows that this is indeed the case. We do not introduce this as evidence that we have correctly identified the gene affected by *n5209* at this point in the narrative. We discuss this experiment in the context of interrogating *cca-1* site of action, but we note that these data further support our interpretation of the mapping data.

*7) Figure 3 legend. The authors write that one copy of n5209 was sufficient to suppress egl-6(gf). I suggest they change the wording to say it was sufficient to partially suppress egl-6(gf). Figure 1 shows that two copies of n5209 appear to fully suppress, and one copy does not cause nearly as much suppression. When I read the unmodified "suppress", I feel that full suppression is implied, and that simply saying "suppress" is not accurate here, especially since the authors emphasize the weaker "partial suppression" shown in Figure 3.*

We agree with the reviewers’ reasoning and have made the change (Figure 3 legend).

*8) Subsection “n5209 affects the T-type calcium channel CCA-1”, last sentence. "n5209 does not cause loss of cca-1 function". I feel this concluding statement is an oversimplification of the results. The fact is that the deletion mutation does weakly partially suppress egl-6(gf), and n5209/+ and n5209/n5209 cause stronger suppression of egl-6(gf). So, the n5209 phenotype shares some similarity with the null phenotype and n5209 may have some loss of function character. I'd be fine with saying "n5209 does not simply cause loss of cca-1 function."*

We agree, and we have changed this text (subsection **“***n5209* affects the T-type calcium channel CCA-1”, last sentence).

*9) In the subsection “n5209 alters the voltage dependency of the voltage-gated T-type calcium channel”, the authors put a P to S point mutation into a mammalian ortholog of cca-1 analogous to the n5209 P to S mutation in cca-1, and characterize effects of the mutation on mammalian channel activity. It is implied, but not stated explicitly, that the authors either didn't try or tried and failed to get C. elegans CCA-1 functionally expressed in Xenopus oocytes, decided to functionally characterize the mammalian channel and its mutant instead, and that they are inferring that the P to S point mutation has similar effects on the worm and mammalian channels. There is nothing wrong with the authors' reasoning, but instead of making the readers read between the lines and infer all this, I would prefer it if the authors simply said these things explicitly. Perhaps this is a subjective matter of style, but I prefer things said out in the open so that reading the paper is less of a puzzle for the reader to solve.*

We failed to functionally express CCA-1 in *Xenopus oocytes*. Fortunately, the sequence affected by *n5209* is almost perfectly conserved between CCA-1 and its mammalian homolog, which is readily expressed in *Xenopus oocytes*. We apologize for the lack of clarity with respect to why we studied the effect of the *n5209* mutation on a vertebrate homolog. We have changed the text to better explain this experiment (subsection “*n5209* alters the voltage dependency of the voltage-gated T-type calcium channel”).

*10) A separate technical concern on Figure 6: The authors show a N value for the number of eggs staged. They do not indicate (as far as I can find) whether all the eggs assayed in each Figure panel came from a single transgenic line, or from multiple lines. Because there is line to line variation in expression levels of C. elegans extrachromosomal transgenes, it would be better if the authors analyzed multiple lines, and then either pool the data from several lines, tell us they are showing data from a representative line, or if they are picking their "best" line, tell us that. In any case, some more explicit information on how the experiment was done is in order.*

The data listed in Figure 6 are from a representative transgenic line, but we have collected data from other lines made with the same RNAi constructs. We have made more explicit that the data in this figure come from a single transgenic line (Figure 6 legend). To show that the line is indeed representative we are including data from the other lines that we tested in an addendum for reviewers (see Figure 9–Figure 11). We did not feel that these data would add value to the manuscript, but we agree that it is an important technical consideration.

Author response image 2.**DOI:**
http://dx.doi.org/10.7554/eLife.22771.030

Author response image 3.**DOI:**
http://dx.doi.org/10.7554/eLife.22771.031

Author response image 4.**DOI:**
http://dx.doi.org/10.7554/eLife.22771.032

*11) The Discussion is largely speculative, although speculation is appropriate and welcome in a Discussion. The first section (subsection “Mechanisms by which T-type currents might gate inhibitory peptidergic modulation”) speculates about how retuned T type channels affect the HSN, but because the HSN is not so far accessible to direct electrophysiological recording, the ideas explored remain untestable.*

We agree. As of yet, neither we nor others have been able to analyze HSNs using electrophysiological recordings. However, given our calcium imaging data and the very clear literature linking T-type channels to control of resting potential and excitability, we feel that our discussion of these points is reasonable. We agree that there are many exciting experiments waiting to be done, especially with respect to the question of whether certain modulatory systems are preferentially affected by certain biophysical mechanisms that regulate excitability and resting potential.

*The second section (subsection “How T-type channels alter the activity of HSNs suggests an integrative mechanism in targets receiving serotonergic input”) analyzes results of the calcium imaging. Because the calcium imaging was done with immobilized animals on agarose pads, a condition in which "normal" egg laying behavior with active and inactive phases cannot occur and in which HSN activity is not correlated with egg laying behavior, there are limits to how well one can interpret what the data mean for normal egg laying behavior.*

This comment refers to the fact that we performed our experiments on immobilized animals, and it is possible that different patterns of activity might be observed in freely behaving animals. We agree. Since we submitted our manuscript, Collins, Koelle and colleagues have published an elegant analysis of activity in the egg-laying system of unrestrained animals (Collins et al.*, eLife* 2016). Importantly, neural activity that is uncorrelated with behavior is also seen in unrestrained animals, and we have tailored this part of our Discussion to describe the similarities between the activity we recorded in restrained animals and what is seen in unrestrained animals. We have also made clear that under the experimental conditions we used it is rare to see egg-laying events. These changes are in the first paragraph of the subsection “How T-type channels alter the activity of HSNs suggests an integrative mechanism in targets receiving serotonergic input”.

*The final Discussion section hypothesizes that T type Ca^2+^ channels generally "gate" neuromodulation and therefore could be good drug targets for psychiatric diseases that seem to involve neuromodulation. This is an interesting idea, but I am not clear how one would achieve a very specific effect with such drugs since T type channels are presumably expressed very widely in the brain.*

This is a fascinating question. T-type channels are expressed throughout the mammalian brain, and we show that they are also expressed widely in the nervous system of *C. elegans*, where we see a very specific effect of a mutation on reproductive behavior. This is the basis for our speculation. We also note that several therapeutics indicated for specific affective or cognitive disorders act on widespread targets (lithium nicely treats mania and is thought to act on voltage-gated sodium channels; electroconvulsive therapy is used to treat major depression that is refractory to selective serotonin reuptake inhibitors). But we agree with the reviewers’ suggestion and have altered the text to better address this point (subsection “T-type calcium channels might play a widespread role in gating the response to modulatory inputs to neural circuits”, last paragraph).

*12) In Figure 1, it was not clear how the timing parameters for egg-laying were determined (the Methods says using MATLAB scripts, but it is not clear how these work).*

We apologize for this, and we have clarified the Methods (subsection “Behavior assays”).

*Also, I'm not sure the plot in Figure 1 illustrates the phenotype of the mutant very well; a histogram of interval times on a log scale would probably work better.*

We have now included this as Figure 1—figure supplement 2, and referred to this alternate representation of the data in the **subsection “***n5209* does not change the temporal structure of female reproductive behavior”and in the legend for Figure 1.

*13) I think it would have been useful for the authors to try to relate the behavioral phenotype to the neural phenotype seen in HSN calcium imaging. For instance, recent papers have linked individual egg-laying events to large calcium spikes in HSN, so perhaps the fact that the cca-1 mutations cause high frequency but low amplitude calcium activity in HSN is consistent with the lower rate of egg-laying within clusters that they observe behaviorally.*

*14) In the calcium imaging experiments (Figure 7), were any egg-laying events observed? If so it would be interesting to see how they correlate with the calcium traces; if none were seen this should be noted.*

Unfortunately, we only observed one egg-laying event during our recordings, perhaps as a result of our recording conditions. As noted above in point #11, recent published work from Collins, Koelle and colleagues show that HSNs are active even when the animal is not laying eggs. We have related our observations to these published data in the Discussion (subsection “How T-type channels alter the activity of HSNs suggests an integrative mechanism in targets receiving serotonergic input”, first paragraph).